# DVLA-RL: Dual-Level Vision-Language Alignment with Reinforcement Learning Gating for Few-Shot Learning

**Wenhao Li[1,2]\*, Xianjing Meng[3]\*, Qiangchang Wang[1]†, Zhongyi Han[1], Zhibin Wu[1], Yilong Yin[1]†**
[1]Software School, Shandong University    [2]Shenzhen Loop Area Institute
[3]School of Computing and Artificial Intelligence, Shandong University of Finance and Economics
{wenhao.li, zhibinwu}@mail.sdu.edu.cn   rongmengyuan@gmail.com
{qiangchang.wang, zhongyi.han, ylyin}@sdu.edu.cn

## Abstract

Few-shot learning (FSL) aims to generalize to novel categories with only a few samples. Recent approaches incorporate large language models (LLMs) to enrich visual representations with semantic embeddings derived from class names. However, they overlook progressive and adaptive alignment between vision and language from low-level to high-level semantics, resulting in limited semantic gains. To address these challenges, we propose Dual-level Vision-Language Alignment with Reinforcement Learning gating (DVLA-RL), which consists of Dual-level Semantic Construction (DSC) and RL-gated Attention (RLA). Specifically, DSC conditions LLMs on both class names and support samples to generate discriminative attributes, progressively selects the most relevant ones, and then synthesizes them into coherent class descriptions. This process provides complementary low-level attributes and high-level descriptions, enabling both fine-grained grounding and holistic class understanding. To dynamically integrate dual-level semantics along with the visual network layers, RLA formulates cross-modal fusion as a sequential decision process. A lightweight policy trained with episodic REINFORCE adaptively adjusts the contributions of self-attention and cross-attention to integrate textual and visual tokens. As a result, shallow layers refine local attributes and deep layers emphasize global semantics, enabling more precise cross-modal alignment. This achieves class-specific discrimination and generalized representations with merely a few support samples. DVLA-RL achieves new state-of-the-art performance across nine benchmarks in three diverse FSL scenarios.

## 1 Introduction

Deep learning has achieved remarkable success, driven by large-scale labeled datasets(He et al., 2016; Dosovitskiy et al., 2021; Wang et al., 2022; 2025b; Liang et al., 2025; Wang et al., 2023a; Meng et al., 2024; Li et al., 2025a; Wang et al., 2025a; Sun et al., 2025; Ma et al., 2025a). However, acquiring such annotations is often costly or even impractical in real-world settings. To reduce the reliance on abundant labeled data and simulate human rapid learning ability from limited experience, Few-Shot Learning (FSL) (Vinyals et al., 2016) is proposed to generalize the knowledge learned from a base set to novel and unseen tasks using a few labeled samples. FSL has garnered significant attention due to its broad application prospects in real-world scenarios, such as rare disease diagnosis and industrial anomaly detection.

FSL is typically formalized as an $N$-way $K$-shot episodic task (Snell et al., 2017). In each task, a support set contains $N$ classes with $K$ labeled samples per class, while a query set with several testing samples is involved to evaluate model performance. Most FSL methods project both support and query samples into a shared embedding space and classify queries by comparing distances to supports. Researchers have conducted extensive studies on vision-based approaches (Snell et al.,

---

\*Equal contribution
†Corresponding authors.

2017; Zhang et al., 2020; Dong et al., 2022; Hiller et al., 2022; Sun & Gao, 2023; Qi et al., 2023; Baik et al., 2024; Ye et al., 2024; Kim et al., 2024; Poul et al., 2024; Hu et al., 2024; Guo et al., 2025), aiming to extract class-relevant features from images and reduce intra-class variation. Despite their success, available representations may not be sufficiently discriminative for accurate classification due to limited samples, especially in the 1-shot scenario with only one labeled sample.

Several recent methods (Xing et al., 2019; Xu & Le, 2022; Pan et al., 2024; Zhang et al., 2024a; Li et al., 2025c; Liu et al., 2025; Li et al., 2025b) attempted to incorporate additional semantic information from other modalities, e.g., natural language, to assist in learning new concepts. SP (Chen et al., 2023) encodes class templates with "A photo of $\{CLASS\}$" to generate semantic prototypes for complementing visual prototypes. SemFew (Zhang et al., 2024a) further enhances semantic quality via large language models (LLMs) by expanding class names into textual descriptions with high-level information. However, they often ignore the low-level discriminative patterns that are critical for extracting class-specific features. An alternative (Liu et al., 2025) attempted to generate detailed class entities to replace high-level semantics. However, these methods rely solely on low-level or high-level semantic embeddings, failing to progressively align visual features at different layers with corresponding levels of semantics. Moreover, their static fusion modules struggle to perform adaptive vision-language alignment across layers, resulting in limited semantic gains.

To address these limitations, a Dual-level Vision-Language Alignment with Reinforcement Learning gating (DVLA-RL) is proposed to achieve hierarchical and dynamic cross-modal alignment from low-level to high-level semantics. Specifically, DVLA-RL consists of Dual-level Semantic Construction (DSC) and the RL-gated Attention (RLA). In DSC, an LLM is conditioned on both class names and support samples to generate fine-grained attribute candidates. A progressive Top-k selection strategy then iteratively filters these attributes by measuring their semantic relevance to retain only the most discriminative ones, suppressing semantic hallucinations from irrelevant attributes. The selected attributes are further synthesized into coherent class descriptions, providing complementary local and global semantic guidance. To fully integrate these dual-level cues into visual feature extraction, RLA formulates cross-modal fusion as a sequential decision process. A lightweight policy trained with episodic REINFORCE dynamically balances self-attention and cross-attention between visual and textual tokens across network layers. This adaptive mechanism progressively enables shallow layers to focus on fine-grained local details, while deeper layers emphasize holistic contextual semantics. Consequently, DVLA-RL effectively captures complex cross-modal relationships, enhancing both class-specific discrimination and generalization with merely a few support samples.

Overall, our contributions are summarized as follows:

- A DVLA-RL framework is proposed to achieve hierarchical and dynamic visual-language alignment between low-level and high-level feature extraction.

- A DSC module is proposed to consistently generate fine-grained attributes and coherent descriptions as complementary semantics, effectively alleviating semantic hallucinations.

- An RLA module is proposed to dynamically balance self-attention and cross-attention between vision and language tokens through reinforcement learning across network layers.

- Extensive experiments are conducted on nine popular benchmark datasets across three distinct FSL scenarios, indicating the superior performance over the state-of-the-art methods.

## 2 RELATED WORK

**Visual-based Methods.** Few-shot learning (FSL) has attracted extensive attention due to its wide application potential in real-world scenarios. Visual-based methods focus on purely visual approaches that learn class-relevant representations from images. They are divided into two types. On the one hand, optimization-based methods, such as MAML (Finn et al., 2017) and NIW-Meta (Kim et al., 2024), aim to quickly adapt to novel tasks with only a few gradient steps. However, full-model adaptation under limited samples is prone to instability and degraded generalization. On the other hand, metric-based methods focus on learning discriminative feature embeddings where query samples can be classified by similarity to class prototypes. Representative works include MatchingNet (Vinyals et al., 2016) using cosine similarity and ProtoNet (Snell et al., 2017) using Euclidean distance, with numerous variants exploring different backbones, distance metrics, and

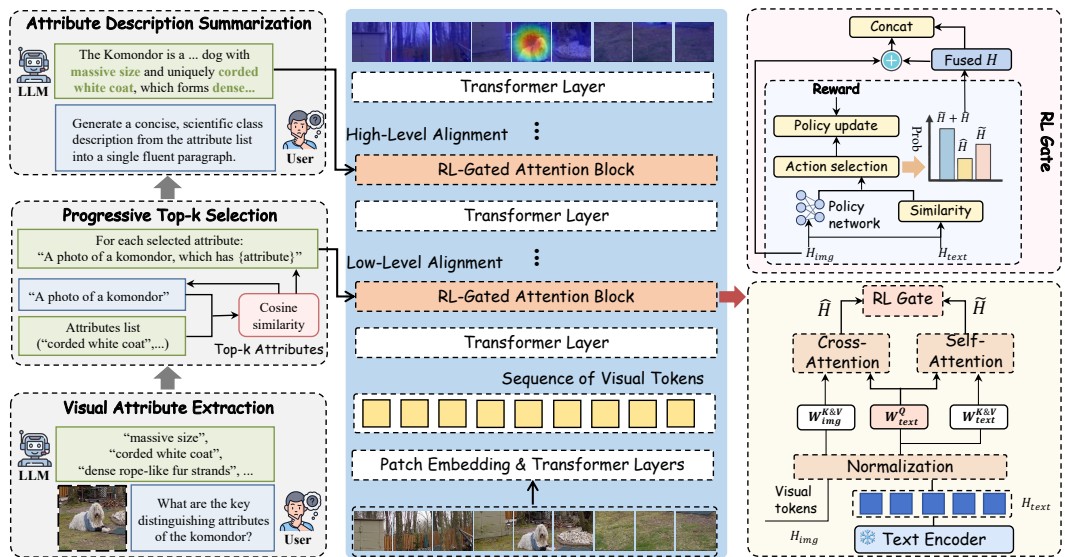

Figure 1: Overview of the proposed DVLA-RL framework. Dual-level Semantic Construction (DSC) extracts visual attributes with LLMs, progressively selects the most discriminative ones, and synthesizes them into class descriptions. Adaptive RL-Gated Attention (RLA) integrates these dual-level semantics with visual tokens, dynamically balancing self- and cross-attention between visual and textual tokens across layers for hierarchical and adaptive vision-language alignment.

prototype designs (Zhang et al., 2020; Dong et al., 2022; Hiller et al., 2022; Sun & Gao, 2023; Du et al., 2023; Hao et al., 2023; Baik et al., 2024; Kim et al., 2024; Poul et al., 2024; Hu et al., 2024).

**Semantic-based Methods.** Relying solely on limited visual samples makes it difficult to acquire discriminative features. To mitigate this, semantic-based methods (Xing et al., 2019; Xu & Le, 2022; Chen et al., 2023; Li et al., 2024; Pan et al., 2024; Zhang et al., 2024a; Liu et al., 2025) incorporate additional semantic information from language modality to complement limited visual evidence. Early AM3 (Xing et al., 2019) incorporates class-level semantics into visual prototypes. KTPP (Li et al., 2024) introduces pyramid-shaped cross-modal prompts to guide support features. SIFT (Pan et al., 2024) further employs category-specific semantic embeddings to generate enriched features through an encoding-decoding process. Recent SemFew (Zhang et al., 2024a) and ECER (Liu et al., 2025) explore the use of large language models (LLMs) to construct global abstractions and local structures, but both rely on single-level semantics and static MLP-based fusion modules that lack adaptivity across network depths during visual extraction. In contrast, DVLA-RL achieves hierarchical and dynamic cross-modal alignment by progressively coordinating low-level attributes and high-level descriptions with visual features through an adaptive RL-gated attention mechanism. To the best of our knowledge, this is the first attempt to introduce reinforcement learning for vision-language alignment in few-shot learning.

## 3 METHODOLOGY

Fig. 1 shows a whole overview of DVLA-RL framework. In this section, two novel components of Dual-level Semantic Construction (DSC) and Adaptive RL-Gated Attention (RLA) are detailed.

### 3.1 DUAL-LEVEL SEMANTIC CONSTRUCTION

**Visual Attribute Extraction.** Given a support category $C_{sup}^i$ and its support samples, the first step is to discover discriminative attributes that highlight inter-class differences. For example, *corded white coat"* or *massive size"* can differentiate the Komondor from other breeds. To obtain such cues, we query large language models (LLMs) with the prompt *["What are the key distinguishing attributes of the CLASS in the given image? List concise attributes"]*, which yields a set of candidate attributes:

$$A^{C_{sup}^i} = \mathcal{L}_\theta\big(P_{dis}(C_{sup}^i)\big), \tag{1}$$

where $A^{C_{sup}^i} = \{a_1, \ldots, a_s\}$ denotes the attribute set, $\mathcal{L}_\theta$ is the LLM, and $P_{dis}$ is the attribute discovery prompt.

**Progressive Top-$k$ Selection.** Not all generated attributes are equally relevant. To refine them, each attribute $a_j$ is encoded by the CLIP text encoder and scored against the current template embedding using cosine similarity as follows:

$$s_j = \cos\big(T^{(i)}, a_j\big), \tag{2}$$

where $T^{(i)}$ is the evolving template at iteration $i$ and $T^{(0)} = [$“A photo of a {CLASS}”$]$. $s_j$ denotes the similarity score. At each step, the most relevant attribute is appended to the template and used to update $T^{(i+1)}$. This progressive refinement continues until $k$ attributes are retained as follows:

$$\hat{A}^{C_{sup}^i} = \text{Prog-Top-}k\big(\{(a_j, s_j)\}_{j=1}^s\big). \tag{3}$$

The iterative enrichment ensures that only the most discriminative attributes are preserved, while hallucinated or redundant ones are suppressed. Each selected attribute is then embedded into a sentence template such as *“A photo of a {CLASS}, which has {attribute}”*, providing structured semantics with fine-grained information for subsequent low-level cross-modal alignment.

**Attribute Description Summarization.** To further capture holistic semantics consistent with local details, the selected attributes $\hat{A}^{C_{sup}^i}$ are summarized into a fluent scientific description $D_i$ using an LLM with the prompt *[“Generate a concise scientific class description from the attribute list into a single fluent paragraph”]*. For instance, the Komondor may be described as: *“The Komondor is a ... dog with massive size and uniquely corded white coat, which forms dense...”* Formally:

$$D_i = \mathcal{L}_\theta\big(P_{sum}(\hat{A}^{C_{sup}^i})\big), \tag{4}$$

where $P_{sum}$ is the summarization prompt. These global descriptions complement local attributes, resulting in dual-level semantics from low-level to high-level, jointly enhancing visual features.

## 3.2 Adaptive RL-gated Attention

Given visual tokens $H_{\text{img}}$ and textual semantics $H_{\text{text}}$, we first normalize both modalities with a shared operator to harmonize statistics: $\bar{H}_{\text{img}} = \text{Norm}(H_{\text{img}})$ and $\bar{H}_{\text{text}} = \text{Norm}(H_{\text{text}})$. RLA then applies two reciprocal attention operators. In the image-guided path, textual queries $Q$ attend to visual keys $K$ and values $V$ by cross-attention, leading to a visually grounded representation that retrieves discriminative and class-specific image regions conditioned on textual semantics. In the text-guided path, $Q, K, V$ originate from textual tokens, resulting in a textually grounded refinement of semantic relations. Both paths follow the standard scaled dot-product attention. Formally:

$$\hat{H} = \text{Attn}\big(W_{\text{text}}^q \bar{H}_{\text{text}}, W_{\text{img}}^k \bar{H}_{\text{img}}, W_{\text{img}}^v \bar{H}_{\text{img}}\big), \tag{5}$$

$$\tilde{H} = \text{Attn}\big(W_{\text{text}}^q \bar{H}_{\text{text}}, W_{\text{text}}^k \bar{H}_{\text{text}}, W_{\text{text}}^v \bar{H}_{\text{text}}\big), \tag{6}$$

$$\text{Attn}(Q, K, V) = \text{softmax}\big(QK^\top/\sqrt{d}\big)V, \tag{7}$$

where $W^{q/k/v}$ are linear projections into a $d$-dimensional hidden shared space. To adaptively balance the two cross-modal paths, RLA fuses their outputs with a stochastic gate:

$$H = \alpha\,\hat{H} + (1-\alpha)\,\tilde{H}, \qquad \alpha \sim \pi_\theta(\cdot \mid s), \tag{8}$$

$$s = \phi\Big(\big[\text{GAP}(\bar{H}_{\text{img}}) \,\|\, \text{GAP}(\bar{H}_{\text{text}}) \,\|\, \cos(\text{GAP}(\bar{H}_{\text{img}}), \text{GAP}(\bar{H}_{\text{text}}))\big]\Big), \tag{9}$$

$$\pi_\theta(\alpha \mid s) = \text{Beta}\big(\kappa\,p_\theta(s), \ \kappa\,(1 - p_\theta(s))\big), \tag{10}$$

where $\alpha \in [0, 1]$ is the mixing weight between the image-guided output $\hat{H}$ and the text-guided output $\tilde{H}$; $s$ is a compact state summarizing the cross-modal context via global average pooling (GAP) and its cosine similarity; $\phi$ is a lightweight MLP producing $p_\theta(s) \in (0, 1)$; $\pi_\theta$ is a Beta policy whose mean equals $p_\theta(s)$ and concentration $\kappa > 0$ controls exploration versus determinism.

The gate is trained with REINFORCE using a task-aware reward:

$$R_t = \lambda_{\text{sim}} \cdot \cos\big(U\,\text{GAP}(H), \mathbf{t}^\star\big) + \lambda_{\text{imp}} \cdot \big(\text{Acc}_t - \text{Acc}_{t-1}\big), \tag{11}$$

where $\mathbf{t}^\star$ is the 512-dimensional ground-truth text embedding encoded by the CLIP text encoder, and $U$ is a linear projection mapping the fused feature $H$ after GAP to the same text space. The first term promotes visual-text alignment, and the second term measures intra-episode accuracy improvement, with $\mathrm{Acc}_t$ and $\mathrm{Acc}_{t-1}$ denoting query accuracy at the current and previous fusion steps. The weights $\lambda_{\mathrm{sim}}, \lambda_{\mathrm{imp}} > 0$ balance alignment and improvement. The policy gradient is estimated as

$$\nabla_\theta \mathcal{J} = \mathbb{E}\big[(R_t - b_t)\nabla_\theta \log \pi_\theta(\alpha \mid s)\big] \;+\; \tau \, \nabla_\theta \mathsf{H}\big(\pi_\theta(\cdot \mid s)\big), \tag{12}$$

where $b_t$ is an exponential-moving-average baseline for variance reduction and $\tau > 0$ adds an entropy bonus $\mathsf{H}(\cdot)$ to prevent premature collapse.

The fused $H$ is injected back to the backbone via a residual addition followed by concatenation as follows:

$$Output = \mathrm{Concat}\big(H_{\mathrm{img}} + GAP(H), \, H\big), \tag{13}$$

where the global semantic context vector $GAP(H) \in \mathbb{R}^d$ is obtained by averaging all textual tokens and broadcast to all visual tokens for channel-wise visual enhancement. The modulated $H_{img}$ is then concatenated with $H$ to form the extended sequence for subsequent Transformer layers.

For the training objective, the RL loss aggregates per-block contributions as:

$$\mathcal{L}_{\mathrm{RL}} = \sum_{\ell=1}^{L} \left[ -\frac{1}{|\mathcal{Q}|} \sum_{q \in \mathcal{Q}} (R_t - b_t) \log \pi_{\theta_\ell}(\alpha_\ell^{(q)} \mid s_\ell^{(q)}) - \tau \, \frac{1}{|\mathcal{Q}|} \sum_{q \in \mathcal{Q}} \mathsf{H}\big(\pi_{\theta_\ell}(\cdot \mid s_\ell^{(q)})\big) \right], \tag{14}$$

where $\pi_{\theta_\ell}$ is the policy of the $\ell$-th block. During training, $\alpha$ is applied in a soft-hard mixture (sampling for stop-gradient and averaging for differentiable paths), while at inference it is replaced by the expectation $\mathbb{E}_{\pi_\theta}[\alpha]$. Stacking multiple RLA blocks allows the policy to adaptively emphasize attribute-level cues in shallow layers and description-level semantics in deeper layers, enabling hierarchy-aware vision-language alignment in few-shot learning.

Moreover, a prototype classifier is adopted as the training objective. For each class $i$, a prototype embedding $c_i$ is computed by averaging the support features extracted by the backbone $f_\phi$:

$$c_i = \tfrac{1}{K} \sum_{k=1}^{K} f_\phi(x_k). \tag{15}$$

For a query instance $\mathbf{x}^q$, its similarity to each prototype is measured by cosine similarity, and the probability of assigning $\mathbf{x}^q$ to class $i$ is given by:

$$p(y^q = i \mid \mathbf{x}^q) = \frac{\exp\big(\cos(f_\phi(\mathbf{x}^q), c_i)/\tau\big)}{\sum_{j=1}^{N} \exp\big(\cos(f_\phi(\mathbf{x}^q), c_j)/\tau\big)}, \tag{16}$$

where $\tau$ is a temperature parameter and $\cos(\cdot, \cdot)$ denotes the cosine similarity function. The predicted label corresponds to the class with the highest probability. Finally, the overall loss also includes the supervised loss between the probability $p_i$ of the query sample $q$ to the $i$-th class between the probability $p_i$ of the query sample $q$ to the $i$-th class and the corresponding ground-truth label, as follows:

$$\mathcal{L}_{\mathrm{sup}} = -\sum_{q \in \mathcal{Q}} \log p_{y_q^\star}^{(q)}, \tag{17}$$

$$\mathcal{L}_{\mathrm{total}} = \mathcal{L}_{\mathrm{sup}} + \lambda \, \mathcal{L}_{\mathrm{RL}}, \tag{18}$$

where $\lambda$ denotes the trade-off hyperparameter of RL weights. All learnable model parameters are finetuned by minimizing $\mathcal{L}_{\mathrm{total}}$ using episodes randomly sampled from training classes.

## 4    EXPERIMENTS

### 4.1    EXPERIMENTAL DETAILS

**Datasets.**    We evaluate DVLA-RL across three primary tasks of general FSL, fine-grained FSL, and cross-domain FSL. For general FSL, experiments are conducted on three benchmarks, namely mini-ImageNet (Vinyals et al., 2016), *tiered*ImageNet (Ren et al., 2018), and CIFAR-FS (Lee et al., 2019).

Table 1: Results (%) on miniImageNet, *tiered*ImageNet, and CIFAR-FS. The average accuracy with 95% confidence interval is reported. Bold and *Blue* font indicates the best and suboptimal results.

| Model | Venue | miniImageNet | | *tiered*ImageNet | | CIFAR-FS | |
|---|---|---|---|---|---|---|---|
| | | 1-shot | 5-shot | 1-shot | 5-shot | 1-shot | 5-shot |
| ProtoNet (Snell et al., 2017) | NeurIPS | $62.39_{\pm.21}$ | $80.53_{\pm.14}$ | $68.23_{\pm.23}$ | $84.03_{\pm.16}$ | $72.20_{\pm.70}$ | $83.50_{\pm.50}$ |
| AM3 (Xing et al., 2019) | NeurIPS | $65.30_{\pm.49}$ | $78.10_{\pm.36}$ | $69.08_{\pm.47}$ | $82.58_{\pm.31}$ | - | - |
| DeepEMD (Zhang et al., 2020) | CVPR | $65.91_{\pm.82}$ | $82.41_{\pm.56}$ | $71.16_{\pm.87}$ | $86.03_{\pm.58}$ | - | - |
| SUN (Dong et al., 2022) | ECCV | $67.80_{\pm.45}$ | $83.25_{\pm.30}$ | $72.99_{\pm.50}$ | $86.74_{\pm.33}$ | - | - |
| FewTURE (Hiller et al., 2022) | NeurIPS | $68.02_{\pm.88}$ | $84.51_{\pm.53}$ | $72.96_{\pm.92}$ | $86.43_{\pm.67}$ | $72.80_{\pm.88}$ | $86.14_{\pm.64}$ |
| SVAE (Xu & Le, 2022) | CVPR | $74.84_{\pm.23}$ | $83.28_{\pm.40}$ | $76.98_{\pm.65}$ | $85.77_{\pm.50}$ | - | - |
| Meta-AdaM (Sun & Gao, 2023) | NeurIPS | $59.89_{\pm.49}$ | $77.92_{\pm.43}$ | $65.31_{\pm.48}$ | $85.24_{\pm.35}$ | - | - |
| ProtoDiff (Du et al., 2023) | NeurIPS | $66.63_{\pm.21}$ | $83.48_{\pm.15}$ | $72.95_{\pm.24}$ | $85.15_{\pm.18}$ | - | - |
| CPEA (Hao et al., 2023) | ICCV | $71.97_{\pm.65}$ | $87.06_{\pm.38}$ | $76.93_{\pm.70}$ | $90.12_{\pm.45}$ | $77.82_{\pm.66}$ | $88.98_{\pm.45}$ |
| SP (Chen et al., 2023) | CVPR | $72.31_{\pm.40}$ | $83.42_{\pm.30}$ | $78.03_{\pm.46}$ | $88.55_{\pm.32}$ | $82.18_{\pm.40}$ | $88.24_{\pm.32}$ |
| ALFA (Baik et al., 2024) | TPAMI | $66.61_{\pm.28}$ | $81.43_{\pm.25}$ | $70.29_{\pm.40}$ | $86.17_{\pm.35}$ | $76.32_{\pm.43}$ | $86.73_{\pm.31}$ |
| LastShot (Ye et al., 2024) | TPAMI | $67.35_{\pm.20}$ | $82.58_{\pm.14}$ | $72.43_{\pm.23}$ | $85.82_{\pm.16}$ | $76.76_{\pm.21}$ | $87.49_{\pm.12}$ |
| NIW-Meta (Kim et al., 2024) | ICLR | $68.54_{\pm.26}$ | $84.81_{\pm.28}$ | $74.59_{\pm.33}$ | $89.76_{\pm.23}$ | - | - |
| BECLR (Poul et al., 2024) | ICLR | $75.74_{\pm.62}$ | $84.93_{\pm.33}$ | $76.44_{\pm.66}$ | $84.85_{\pm.37}$ | - | - |
| SIFT (Pan et al., 2024) | IJCV | $77.31_{\pm.67}$ | $86.95_{\pm.53}$ | $77.86_{\pm.77}$ | $89.89_{\pm.52}$ | - | - |
| KTPP (Li et al., 2024) | MM | $76.71_{\pm.37}$ | $86.46_{\pm.27}$ | $80.80_{\pm.43}$ | $90.01_{\pm.29}$ | $83.63_{\pm.57}$ | *$90.19_{\pm.30}$* |
| SemFew (Zhang et al., 2024a) | CVPR | $78.94_{\pm.66}$ | $86.49_{\pm.50}$ | *$82.37_{\pm.77}$* | $89.89_{\pm.52}$ | $84.34_{\pm.67}$ | $89.11_{\pm.54}$ |
| UAP (Hu et al., 2024) | NeurIPS | *$81.63_{\pm.28}$* | $79.05_{\pm.19}$ | $79.68_{\pm.30}$ | $76.78_{\pm.21}$ | - | - |
| ECER (Liu et al., 2025) | AAAI | $81.14_{\pm.15}$ | - | $81.81_{\pm.51}$ | - | *$86.01_{\pm.35}$* | - |
| CPL (Guo et al., 2025) | TPAMI | $72.82_{\pm.61}$ | *$87.93_{\pm.37}$* | $78.05_{\pm.70}$ | *$90.89_{\pm.44}$* | $78.82_{\pm.65}$ | $89.98_{\pm.44}$ |
| DVLA-RL | ours | **$81.69_{\pm.36}$** | **$88.25_{\pm.28}$** | **$83.02_{\pm.43}$** | **$91.71_{\pm.29}$** | **$87.18_{\pm.40}$** | **$90.59_{\pm.31}$** |

For fine-grained FSL, we adopt CUB-200-2011 (CUB) (Wah et al., 2011), Stanford Cars (Krause et al., 2013), and Stanford Dogs (Khosla et al., 2011), which contain subtle intra-class variations that pose additional challenges for recognition. In the cross-domain FSL setting, we follow the common protocol (Tseng et al., 2020; Zhou et al., 2023; Zou et al., 2024) by training on miniImageNet and testing on three unrelated domains, including CUB, Places (Zhou et al., 2017), and ChestX (Wang et al., 2017). The detailed dataset statistics regarding the number of categories and images are summarized in the Appendix.

**Training Details.** For training, we adopt a common two-stage FSL strategy (Dong et al., 2022), consisting of large-scale pre-training and meta-tuning. Following recent few-shot studies (Li et al., 2024; Chen et al., 2023; Liu et al., 2025), the visual backbone is chosen as ViT-based Visformer-Tiny (Chen et al., 2021). The text encoder is ViT-B/16 CLIP (Radford et al., 2021) with a 512-dimensional embedding space. To provide dual-level semantic supervision, Qwen2.5-VL-32B (Bai et al., 2025) is employed to generate both attribute-level and description-level textual information. All input images are resized to $224 \times 224$ (Zhang et al., 2024a). The optimization is performed using AdamW (Loshchilov & Hutter, 2017) with an initial learning rate of $5 \times 10^{-4}$ and a cosine learning rate scheduler (Loshchilov & Hutter, 2016). Pre-training is run for 300 epochs on tieredImageNet and 800 epochs on other datasets with a batch size of 512, followed by 100 epochs of episodic meta-tuning. Beta concentration $\kappa = 10$ in RLA while $\lambda_{sim}$ and $\lambda_{imp}$ are set to 0.5 and 1.0, respectively. The reinforcement learning gate uses $\lambda = 0.1$ for balancing the RL objective and $\tau = 0.2$ according to validation accuracy. All experiments are conducted on an NVIDIA RTX 6000 Ada GPU.

**Evaluation Protocol.** For evaluation, we adopt the widely used episodic protocol (Chen et al., 2023; Li et al., 2024; Xing et al., 2019; Li et al., 2020; Xu & Le, 2022; Zhang et al., 2023) in FSL. Specifically, 2000 classification tasks are uniformly sampled from the novel classes that do not overlap with training categories. Each task follows the standard N-way K-shot setting, where 15 query samples per class are included for evaluation. The final performance is reported as the mean classification accuracy across all sampled tasks, along with the 95% confidence interval.

## 4.2 COMPARISON WITH THE SOTA METHODS

**General Few-shot Classification.** Table 1 reports results on miniImageNet, tieredImageNet, and CIFAR-FS. We highlight three observations. (1) Semantic-based methods (e.g., AM3, SP, SIFT, SemFew, and ECER) generally outperform pure metric- or optimization-based approaches, showing the advantage of incorporating language priors into few-shot learning. (2) DVLA-RL consistently

Table 2: Results (%) on fine-grained CUB, Dogs, and Cars. The average accuracy with 95% confidence interval is reported. Bold and *Blue* font indicates the best and suboptimal results.

| Model | Venue | CUB-200-2011 | | Stanford-Dogs | | Stanford-Cars | |
| --- | --- | --- | --- | --- | --- | --- | --- |
| | | 1-shot | 5-shot | 1-shot | 5-shot | 1-shot | 5-shot |
| ProtoNet (Du et al., 2023) | NeurIPS | $63.44_{\pm.56}$ | $83.17_{\pm.35}$ | $41.61_{\pm.50}$ | $76.78_{\pm.36}$ | $45.01_{\pm.49}$ | $87.19_{\pm.31}$ |
| FRN (Wertheimer et al., 2021) | CVPR | $83.55_{\pm.19}$ | $92.92_{\pm.10}$ | $49.37_{\pm.20}$ | $67.13_{\pm.17}$ | $58.90_{\pm.22}$ | $79.65_{\pm.15}$ |
| DAN (Xu et al., 2022) | AAAI | $72.89_{\pm.50}$ | $86.60_{\pm.31}$ | $59.81_{\pm.50}$ | $77.19_{\pm.35}$ | $70.21_{\pm.50}$ | $85.55_{\pm.31}$ |
| MFGN (Yu et al., 2022) | IJCAI | $84.01_{\pm.39}$ | $84.01_{\pm.39}$ | $74.81_{\pm.44}$ | $86.52_{\pm.26}$ | - | - |
| TDM (Lee et al., 2022) | CVPR | $84.36_{\pm.19}$ | $93.37_{\pm.10}$ | $57.64_{\pm.22}$ | $75.03_{\pm.16}$ | $68.36_{\pm.22}$ | $86.14_{\pm.13}$ |
| BSFA (Zha et al., 2023) | TCSVT | $86.00_{\pm.41}$ | $92.53_{\pm.23}$ | $69.58_{\pm.50}$ | $82.59_{\pm.33}$ | $88.93_{\pm.38}$ | $95.20_{\pm.20}$ |
| MLI (Zhao et al., 2024) | TIP | $85.94_{\pm.42}$ | $93.50_{\pm.29}$ | $76.32_{\pm.47}$ | $88.25_{\pm.27}$ | - | - |
| C2-Net (Ma et al., 2024) | AAAI | $83.31_{\pm.41}$ | $92.18_{\pm.23}$ | $75.50_{\pm.49}$ | $87.65_{\pm.28}$ | $88.96_{\pm.37}$ | $95.16_{\pm.20}$ |
| SUITED (Ma et al., 2025b) | AAAI | *$86.02_{\pm.47}$* | *$94.13_{\pm.24}$* | *$76.55_{\pm.47}$* | *$88.86_{\pm.27}$* | *$89.97_{\pm.36}$* | *$96.53_{\pm.16}$* |
| DVLA-RL | ours | **$91.93_{\pm.28}$** | **$95.06_{\pm.19}$** | **$89.64_{\pm.30}$** | **$91.42_{\pm.25}$** | **$92.95_{\pm.24}$** | **$96.59_{\pm.15}$** |

Table 3: Results (%) on cross-domain miniImageNet →CUB, Places, and ChestX. The average accuracy with 95% confidence interval is reported. Bold and *Blue* font indicates the best and suboptimal results.

| Model | Venue | CUB | | Places | | ChestX | |
| --- | --- | --- | --- | --- | --- | --- | --- |
| | | 1-shot | 5-shot | 1-shot | 5-shot | 1-shot | 5-shot |
| GNN (Garcia & Bruna, 2018) | ICLR | $44.40_{\pm.68}$ | $62.87_{\pm.65}$ | $52.42_{\pm.80}$ | $70.91_{\pm.65}$ | $22.00_{\pm.46}$ | $25.27_{\pm.59}$ |
| FWT (Tseng et al., 2020) | ICLR | $45.50_{\pm.46}$ | $64.97_{\pm.68}$ | $53.44_{\pm.79}$ | $70.70_{\pm.67}$ | $22.04_{\pm.44}$ | $25.18_{\pm.45}$ |
| AFA (Hu & Ma, 2022) | ECCV | $46.86_{\pm.70}$ | $68.25_{\pm.65}$ | $54.04_{\pm.75}$ | $76.21_{\pm.60}$ | $22.92_{\pm.20}$ | $25.02_{\pm.20}$ |
| UCD (Oh et al., 2022) | NeurIPS | $40.65_{\pm.68}$ | $58.54_{\pm.70}$ | $51.84_{\pm.72}$ | $72.19_{\pm.60}$ | $22.64_{\pm.40}$ | $26.26_{\pm.45}$ |
| ATA (Wang et al., 2023b) | AIJ | $45.00_{\pm.50}$ | $66.22_{\pm.50}$ | $53.57_{\pm.50}$ | $75.48_{\pm.40}$ | $22.10_{\pm.20}$ | $24.32_{\pm.40}$ |
| LDP-net (Zhou et al., 2023) | CVPR | $49.82_{\pm.70}$ | $70.39_{\pm.66}$ | $53.82_{\pm.71}$ | $72.90_{\pm.63}$ | $23.01_{\pm.45}$ | $26.67_{\pm.40}$ |
| StyleAdv (Fu et al., 2023) | CVPR | $48.49_{\pm.72}$ | $68.72_{\pm.67}$ | $58.58_{\pm.83}$ | $77.73_{\pm.62}$ | $22.64_{\pm.35}$ | $26.07_{\pm.37}$ |
| FAP (Zhang et al., 2024b) | IJCAI | $50.56_{\pm.73}$ | $64.17_{\pm.69}$ | $57.34_{\pm.72}$ | $72.05_{\pm.60}$ | $21.56_{\pm.20}$ | $24.15_{\pm.20}$ |
| FLoR (Zou et al., 2024) | CVPR | $49.99_{\pm.68}$ | $70.39_{\pm.67}$ | $53.18_{\pm.70}$ | $72.31_{\pm.62}$ | $23.11_{\pm.45}$ | $26.70_{\pm.40}$ |
| MEFP (Zhou et al., 2024) | NeurIPS | *$51.55_{\pm.70}$* | *$73.61_{\pm.66}$* | $52.06_{\pm.69}$ | $73.78_{\pm.61}$ | $22.82_{\pm.45}$ | $26.53_{\pm.40}$ |
| SVasP (Li et al., 2025d) | AAAI | $49.49_{\pm.72}$ | $68.95_{\pm.66}$ | *$59.07_{\pm.81}$* | *$77.78_{\pm.62}$* | *$23.23_{\pm.35}$* | *$26.87_{\pm.38}$* |
| DVLA-RL | ours | **$67.46_{\pm.47}$** | **$78.99_{\pm.35}$** | **$69.26_{\pm.45}$** | **$80.70_{\pm.36}$** | **$23.47_{\pm.20}$** | **$26.94_{\pm.22}$** |

achieves the best or second-best accuracy across all datasets and both 1-shot and 5-shot settings. For instance, it obtains 81.69%/88.25% on miniImageNet and 87.18%/90.59% on CIFAR-FS, surpassing the strong baseline SemFew by 0.6%-2.8%. (3) These results indicate that our dual-level semantic construction effectively complements fine-grained attributes with global descriptions, while the RL-gated attention adaptively balances cross-modal fusion across network depths. Such hierarchical alignment allows DVLA-RL to suppress semantic hallucinations and better capture discriminative cues from limited samples, ultimately leading to superior generalization in FSL.

**Fine-grained Few-shot Classification.** The classification results on three fine-grained datasets are presented in Table 2. It can be observed that DVLA-RL achieves the best performance on all benchmarks, significantly surpassing the second-best SUITED by 5.4%-15.3% in the 1-shot setting. In particular, DVLA-RL reaches 91.93%/95.06% on CUB, 89.64%/91.42% on Dogs, and 92.95%/96.51% on Cars, consistently outperforming existing state-of-the-art methods. These results demonstrate that DVLA-RL effectively captures subtle inter-class differences and preserves intra-class consistency by leveraging dual-level semantic construction and adaptive RL-gated attention fusion, thus exhibiting strong generalization ability in challenging fine-grained FSL tasks.

**Cross-Domain Few-Shot Classification.** As shown in Table 3, DVLA-RL consistently outperforms all baselines across both 1-shot and 5-shot tasks on CUB, Places, and ChestX. In particular, it achieves 67.46%/78.99% on CUB and 69.26%/80.70% on Places, exceeding the second-best methods by 2.1%-7.2% under the 1-shot setting. Even on the highly challenging ChestX dataset, DVLA-RL still attains competitive improvements over prior approaches. These results demonstrate that DVLA-RL learns more transferable representations by hierarchically aligning fine-grained attributes and global descriptions, and adaptively balancing cross-modal fusion, thereby generalizing effectively to novel domains under distribution shifts.

Table 4: Ablation study on three datasets under the 1-shot and 5-shot settings. *Attr* and *Desc* means attribute-level and description-level semantics, respectively. *Top-k* means progressive Top-$k$ attribute selection, and *RLA* denotes adaptive RL-gated attention block.

| *Attr* | *Desc* | *Top-k* | *RLA* | miniImageNet | | CIFAR-FS | | CUB | |
|---|---|---|---|---|---|---|---|---|---|
| | | | | 1-shot | 5-shot | 1-shot | 5-shot | 1-shot | 5-shot |
| | | | | $68.67_{\pm.34}$ | $82.97_{\pm.30}$ | $76.67_{\pm.43}$ | $86.89_{\pm.33}$ | $79.47_{\pm.30}$ | $80.63_{\pm.19}$ |
| ✓ | | | ✓ | $75.73_{\pm.36}$ | $85.76_{\pm.28}$ | $84.82_{\pm.39}$ | $88.31_{\pm.30}$ | $88.63_{\pm.31}$ | $90.00_{\pm.20}$ |
| ✓ | | ✓ | ✓ | $76.56_{\pm.35}$ | $85.25_{\pm.29}$ | $85.74_{\pm.39}$ | $88.91_{\pm.30}$ | $89.33_{\pm.30}$ | $90.91_{\pm.21}$ |
| | ✓ | | ✓ | $78.36_{\pm.40}$ | $85.72_{\pm.32}$ | $85.57_{\pm.41}$ | $89.04_{\pm.31}$ | $88.56_{\pm.30}$ | $89.72_{\pm.30}$ |
| ✓ | ✓ | ✓ | ✓ | $\mathbf{81.69}_{\pm\mathbf{.36}}$ | $\mathbf{88.25}_{\pm\mathbf{.28}}$ | $\mathbf{87.18}_{\pm\mathbf{.40}}$ | $\mathbf{90.59}_{\pm\mathbf{.31}}$ | $\mathbf{91.93}_{\pm\mathbf{.28}}$ | $\mathbf{95.06}_{\pm\mathbf{.19}}$ |

## 4.3 MODEL ANALYSIS

**Ablation Study.** We conduct an ablation study on three datasets to evaluate the effectiveness of each component in DVLA-RL, as shown in Table 4. It is observed that (1) Using only attribute-level semantics provides consistent gains over the baseline, verifying that fine-grained attributes can serve as effective discriminative cues. Adding description-level semantics further improves accuracy across datasets, showing the complementarity of local and global guidance. (2) The progressive Top-$k$ selection strategy yields additional improvements (e.g., +1.1% on CUB 1-shot), indicating that iteratively refining the template helps suppress irrelevant attributes and enrich semantic grounding. (3) RLA brings further performance boosts by dynamically balancing self- and cross-attention, especially in the 5-shot setting. (4) The best results are achieved when all components are combined, where DVLA-RL attains 81.69%/88.25% on miniImageNet, 87.18%/90.59% on CIFAR-FS, and 91.93%/95.06% on CUB. These results confirm the necessity of both dual-level semantic construction and adaptive fusion, and highlight the effectiveness of their joint integration.

Table 5: Comparison with different cross-modal fusion and gating methods

(a) Cross-modal fusion mechanisms

| Method | miniImageNet | | CIFAR-FS | |
|---|---|---|---|---|
| | 1-shot | 5-shot | 1-shot | 5-shot |
| SP | 77.81 | 86.01 | 84.53 | 88.21 |
| SemFew | 78.59 | 86.77 | 85.82 | 89.19 |
| ECER | 79.35 | 87.12 | 86.62 | 90.15 |
| ours | **81.69** | **88.25** | **87.18** | **90.59** |

(b) Gating mechanisms

| Method | miniImageNet | | CIFAR-FS | |
|---|---|---|---|---|
| | 1-shot | 5-shot | 1-shot | 5-shot |
| Sum | 79.63 | 86.13 | 85.43 | 88.34 |
| MLP | 80.05 | 86.92 | 85.75 | 89.43 |
| Sigmoid | 80.83 | 87.55 | 86.27 | 89.63 |
| ours | **81.69** | **88.25** | **87.18** | **90.59** |

**Comparison with Fusion and Gating Methods.** As shown in Table 5a, we compare different strategies of cross-modal fusion by replacing our RLA module with representative cross-modal fusion FSL methods. The same set of LLM-generated textual semantics is utilized for all methods. SP employs a simple MLP fusion and performs the weakest due to its limited ability to capture semantic relationships. SemFew and ECER enhance the fusion with larger-scale and multi-level MLP, but their static alignment strategy fails to adapt to varying visual hierarchies, resulting in performance that still falls short of our RLA module. Table 5b shows the replacement of our reinforcement learning gating mechanism with alternative gating mechanisms in RLA. Direct summation ignores task-specific relevance, MLP yields static fusion weights, and Sigmoid MLP provides only marginal nonlinearity. In contrast, our reinforcement learning gate dynamically adjusts the contribution of self-attention and cross-modal attention through reward-guided exploration, leading to consistent improvements.

Table 6: Comparison of computational overhead on tieredImageNet 1-shot tasks

| Method | Params/GFLOPS | Memory (GB) | Semantics (h) | Training (min) | Inference (ms) |
|---|---|---|---|---|---|
| SP (Chen et al., 2023) | $10.8M$ / 1.5 | 4.0 | - | 27 | 88 |
| SemFew (Zhang et al., 2024a) | $19.8M$ / 3.5 | 8.3 | 1.5 | 33 | 107 |
| ECER (Liu et al., 2025) | $26.5M$ / 5.5 | 10.5 | 0.7 | 46 | 121 |
| **ours** | $12.8M$ / 2.0 | 4.8 | 1.0 | **22** | **80** |

**Comparison of computational overhead.** The experiment of computational overhead is conducted on tieredImageNet 1-shot tasks with the same server configuration. The comparison with SP (Chen et al., 2023) (without LLM), SemFew (Zhang et al., 2024a), and ECER (Liu et al., 2025) (with LLMs) is shown in Table 6. The semantic construction stage takes 1.0 h, which is comparable to or lower than existing LLM-based methods. DVLA-RL achieves the shortest training time (22 min) and the lowest inference latency (80 ms). Relative to ECER, DVLA-RL reduces training time by 52% and inference latency by 34%. Moreover, relative to SemFew, DVLA-RL cuts GPU memory consumption from 8.3 G to 4.8 G and reduces inference time by 25%.These advantages arise from the plug-in and lightweight design of DVLA-RL. Texts are generated once offline and used directly in cross-modal fusion, avoiding the extra LLM-based pretraining in ECER. During training/inference, DVLA-RL employs only a lightweight RL gating for fusion rather than the larger MLP module with 7.3M parameters in SemFew.

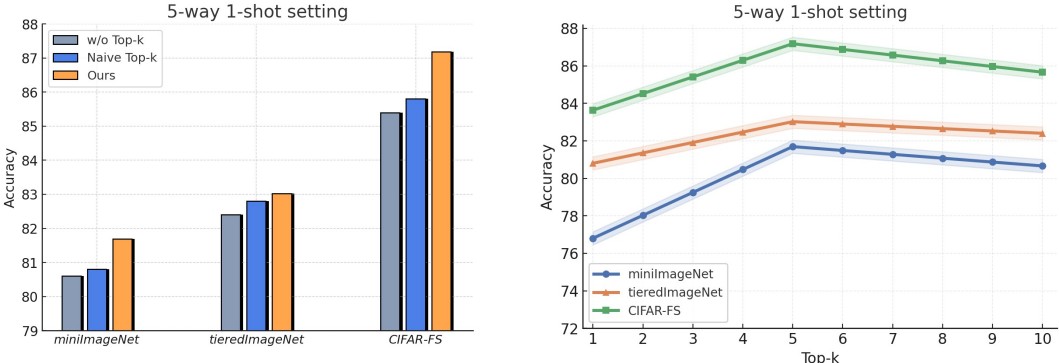

Figure 2: The effect of different selection strategies and Top-$k$ number.

**The Effect of Top-$k$ in DSC.** As shown in Figure 2, we first compare different attribute selection strategies. It can be observed that both the naive Top-$k$ and our progressive Top-$k$ strategy outperform the setting without attribute filtering, verifying the necessity of selecting discriminative attributes. Moreover, our progressive strategy consistently achieves the highest accuracy across datasets, showing that iterative template updating helps retain more relevant semantics. We further study the impact of the Top-$k$ number. Performance improves as $k$ increases, and reaches the peak at around $k = 5$ across datasets, after which accuracy drops when too many attributes are included. This demonstrates that a moderate number of attributes provides sufficient semantic richness, while excessive ones introduce noise that harms generalization.

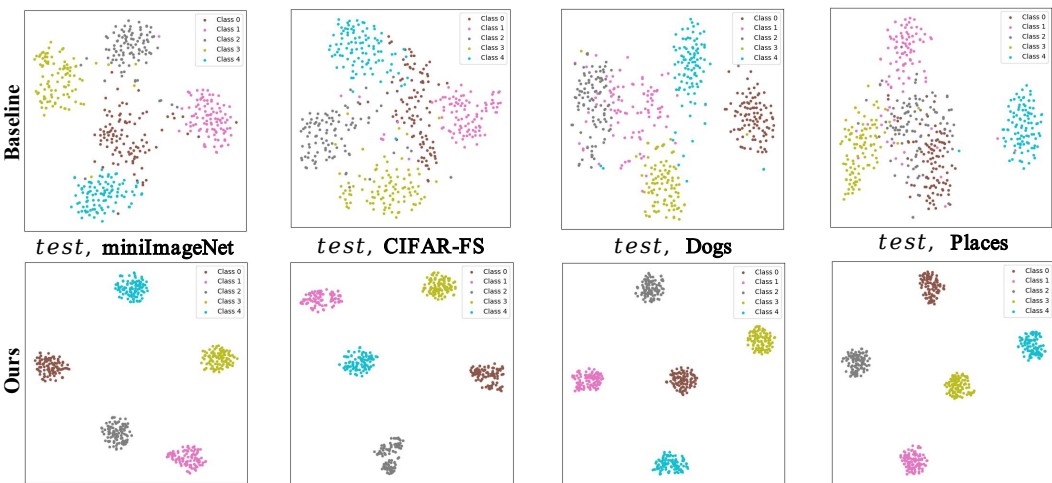

Figure 3: T-SNE visualization on novel classes from four datasets.

**T-SNE Visualization.** We present T-SNE (Maaten & Hinton, 2008) visualizations of novel classes from miniImageNet, CIFAR-FS, Dogs, and Places in Figure 3. T-SNE, as a low-dimensional projection technique, focuses on highlighting differences in overall distribution structures. To more clearly observe the feature distribution effects, 100 support images are randomly sampled from each category. The baseline is constructed by removing our DSC and RLA modules while keeping the same backbone and the same sampled support images to ensure fair comparison. Compared with the baseline, where class clusters are entangled and boundaries are ambiguous, our method produces more compact and well-separated clusters across all datasets. This indicates that dual-level semantic construction provides complementary local and global guidance, while the RL-gated fusion adaptively aligns cross-modal features, resulting in more discriminative and transferable representations. These visualizations intuitively confirm the effectiveness of DVLA-RL in enhancing class separability under diverse scenarios.

Table 7: Comparison with different LLMs for classification and semantic generation on miniImageNet, CIFAR-FS, and CUB. The average accuracy with 95% confidence interval is reported.

| Model | Role | miniImageNet | | CIFAR-FS | | CUB | |
|---|---|---|---|---|---|---|---|
| | | 1-shot | 5-shot | 1-shot | 5-shot | 1-shot | 5-shot |
| Qwen2.5-VL-32B | Classification | $82.41_{\pm.87}$ | $83.22_{\pm.76}$ | $85.98_{\pm.77}$ | $86.92_{\pm.73}$ | $87.85_{\pm.89}$ | $88.50_{\pm.81}$ |
| GPT-4 turbo | Semantics | $81.12_{\pm.35}$ | $\mathbf{88.91}_{\pm\mathbf{.27}}$ | $86.72_{\pm.40}$ | $90.21_{\pm.30}$ | $91.41_{\pm.30}$ | $94.67_{\pm.18}$ |
| GPT-4o | Semantics | $82.04_{\pm.32}$ | $87.79_{\pm.28}$ | $86.61_{\pm.41}$ | $90.07_{\pm.30}$ | $\mathbf{92.45}_{\pm\mathbf{.27}}$ | $94.27_{\pm.20}$ |
| Qwen2.5-VL-32B | Semantics | $81.69_{\pm.36}$ | $88.25_{\pm.28}$ | $\mathbf{87.18}_{\pm\mathbf{.40}}$ | $\mathbf{90.59}_{\pm\mathbf{.31}}$ | $91.93_{\pm.28}$ | $\mathbf{95.06}_{\pm\mathbf{.19}}$ |

**The Effect of Different LLMs.** Table 7 compares Qwen2.5-VL-32B, GPT-4 turbo, and GPT-4o on three datasets. To ensure fairness, the direct classification evaluation strictly follows the same N-way K-shot evaluation protocol in FSL, where frozen Qwen2.5-VL-32B receives all support images and the query image via a unified multimodal prompt *["Given K support examples for N classes as above, please identify which class the query image belongs to"]* and produces a class prediction through generative decoding. However, Qwen2.5-VL-32B performs relatively poorly (83.22% on miniImageNet under 5-shot setting), showing the difficulty of adapting LLMs to few-shot recognition. In contrast, using LLMs for semantic generation substantially improves results. GPT-4o excels in the 1-shot setting (92.45% on CUB), while Qwen2.5-VL-32B leads in 5-shot scenarios (95.06% on CUB). These findings highlight that DVLA-RL is robust across different LLMs, and that LLMs are far more effective as semantic generators than direct classifiers.

## 5 CONCLUSION

In this work, we proposed DVLA-RL, a novel framework for hierarchical and adaptive vision-language alignment in Few-Shot Learning (FSL). The Dual-level Semantic Construction (DSC) module progressively extracts fine-grained attributes and synthesizes coherent class descriptions, offering complementary low-level and high-level semantics. To dynamically integrate these semantics, the RL-gated Attention (RLA) module formulates cross-modal fusion as a sequential decision process, adaptively balancing self- and cross-attention between visual and textual tokens with reinforcement learning across network layers. This design enables shallow layers to capture local details and deeper layers to model global context, leading to more discriminative and generalizable representations. Extensive experiments on nine popular benchmarks across three distinct FSL scenarios demonstrate that our DVLA-RL achieves new state-of-the-art results.

## ACKNOWLEDGMENT

This work was supported in part by the Natural Science Foundation of China under Grant 62176139 and 62406177, in part by the Jinan Science and Technology Project under Grant 202428008, in part by the Shandong Excellent Young Scientists Fund (Oversea) under Grant 2024HWYQ-027, in part by the Natural Science Foundation of Shandong province, China under Grant ZR2023QF124, in part by the Young Scholars Program of Shandong University.

ETHICS STATEMENT

This work adheres to the ICLR Code of Ethics.[1] Our study does not involve human subjects, sensitive personal data, or potentially harmful applications. The proposed method does not introduce risks of misuse, privacy concerns, or discrimination.

REPRODUCIBILITY STATEMENT

We have taken extensive steps to ensure the reproducibility of our work. All datasets used in our experiments are publicly available and referenced in the paper. The implementation details are thoroughly described in the paper. We are committed to ensuring fairness and transparency in both experimentation and reporting. An code link of model training and testing will be released after paper acceptance to facilitate reproducibility of the reported results.

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

# A   APPENDIX

## LLM USAGE STATEMENT

Large language models (LLMs) were used in this work as an assistive tool to improve the clarity and fluency of writing. All technical content, including model development and empirical evaluations, was conceived, implemented, and validated by the authors. The authors take full responsibility for the correctness and integrity of the paper.

## A.1   TRAINING ALGORITHM

---

**Algorithm 1:** Meta-training algorithm of the proposed DVLA-RL.

---

**Input:** Training set $\mathcal{D}$, visual backbone $f_\varphi$, text encoder $g$, LLM $L_\theta$, RL policies $\{\pi_{\theta_\ell}\}_{\ell=1}^L$

**while** *not converged* **do**

   1. Sample an $N$-way $K$-shot task $\mathcal{T} = \{\mathcal{S}, \mathcal{Q}\}$ from $\mathcal{D}$ by the episodic strategy;

   **for** *each class $y_i$ in $\mathcal{S}$* **do**

      2. Query LLM to extract candidate attributes $A^{C_{sup}^i} = \{a_1, \ldots, a_s\}$ according to Eq. 1;

      3. Perform progressive Top-$k$ attributes selection for $\hat{A}^{C_{sup}^i}$ as in Eq. 2- 3 ;

      4. Summarize into a coherent description $D_i$ via LLM as in Eq. 4;

   **for** *each support image $x$ in $\mathcal{S}$* **do**

      5. Extract visual tokens $H_{\text{img}}$ with intermediate layer $f_\varphi$ and textual tokens $H_{\text{text}}$ with $g$;

      **for** *layer $\ell = 1$ to $L$* **do**

         6. Select layer-specific text tokens: attribute for shallow layers, description for deep;

         7. Compute image-guided $\hat{H}$ and text-guided $\tilde{H}$, followed by fusion $H$ via Eq. 5- 8;

         8. Construct state $s_\ell$ and sample gate $\alpha_\ell$ from RL policy $\pi_{\theta_\ell}$ via Eq. 9- 10;

         9. Integrate $H_{\text{img}}^\ell$ with $H$ by residual connection according to Eq. 13;

         10. Compute rewards $R_t$ and the RL loss $\mathcal{L}_{\text{RL}}$ according to Eq. 11, 12, 14;

   10. Construct class prototype $C_i$ by averaging support features via Eq. 15;

   **for** *each query image $x_j$ in $\mathcal{Q}$* **do**

      11. Compute the prediction scores $p_{x_j}$ between $f_\varphi(x_j)$ and prototypes $C_i$ in Eq. 16;

      12. Calculate the cross-entropy loss $\mathcal{L}_{\text{sup}}$ between $p_{x_j}$ and the ground-truth $y_j$ in Eq. 17;

   13. Compute the overall loss $\mathcal{L}_{\text{total}} = \mathcal{L}_{\text{sup}} + \lambda \mathcal{L}_{\text{RL}}$ via Eq. 18;

   14. Update $f_\varphi$ and $\{\theta_\ell\}$ by backpropagation;

**Output:** Meta-trained backbone $f_\varphi$ for testing.

---

We describe the entire training procedure of DVLA-RL in detail in Algorithm 1. The training process follows an episodic meta-learning paradigm and incorporates two key modules: (1) *Dual-level Semantic Construction (DSC)*, which derives complementary low-level attributes and high-level descriptions from LLMs, enabling both fine-grained grounding and holistic class understanding ; and (2) *Adaptive RL-gated Attention (RLA)*, which dynamically fuses image-guided and text-guided paths using reinforcement-learned stochastic gates to achieve hierarchical cross-modal alignment. These modules jointly enable the construction of prototypes that integrate dual-level semantics with discriminative visual features. The complete process, including semantic generation, feature extraction, cross-modal fusion, prototype construction, loss computation, and parameter updates, is formalized in Algorithm 1.

## A.2   DATASETS

We conduct comprehensive evaluations on eight widely adopted few-shot learning benchmarks, encompassing three representative scenarios. For standard few-shot classification, we employ mini-ImageNet, tieredImageNet, and CIFAR-FS. For fine-grained few-shot classification, we adopt CUB-200-2011, Stanford-Cars, and Stanford-Dogs, which require recognizing subtle inter-class variations. For cross-domain few-shot classification, we include Places and ChestX, which assess the robustness

Table 8: The splits of categories and the number of categories/images in each few-shot dataset.

| Dataset | #(Class) | | | #(Image) | |
|---|---|---|---|---|---|
| | $\mathcal{D}_{\text{train}}$ | $\mathcal{D}_{\text{valid}}$ | $\mathcal{D}_{\text{test}}$ | Test | Total |
| miniImageNet (Vinyals et al., 2016) | 64 | 16 | 20 | 12,000 | 60,000 |
| *tiered*ImageNet (Ren et al., 2018) | 351 | 97 | 160 | 206,209 | 779,165 |
| CIFAR-FS (Lee et al., 2019) | 64 | 16 | 20 | 12,000 | 60,000 |
| CUB-200-2011 (Wah et al., 2011) | 100 | 50 | 50 | 2,958 | 11,788 |
| Stanford-Cars (Krause et al., 2013) | 130 | 17 | 49 | 4,103 | 16,185 |
| Stanford-Dogs (Khosla et al., 2011) | 70 | 20 | 30 | 5,115 | 20,580 |
| Places (Zhou et al., 2017) | 183 | 91 | 91 | 18,200 | 73,000 |
| ChestX (Wang et al., 2017) | / | / | 7 | 25,847 | 25,847 |

of models under significant distribution shifts. The detailed category splits and image statistics of all datasets are provided in Table 8.

**Standard Few-Shot Classification.** **miniImageNet** is a subset of ILSVRC-12 containing 100 categories with 600 images per class. We use 64 classes for training, 16 for validation, and 20 for testing. **tieredImageNet** is a larger subset of ILSVRC-12, organized into high-level categories to reduce semantic overlap between training and test classes. It consists of 608 classes in total, with 351/97/160 classes for training/validation/testing, respectively, and over 770,000 images. **CIFAR-FS** is derived from CIFAR-100 and contains 100 classes with 600 images per class. The standard split includes 64 training classes, 16 validation classes, and 20 test classes, each with lower-resolution images ($32 \times 32$), providing a more challenging setting.

**Fine-Grained Few-Shot Classification.** **CUB-200-2011 (CUB)** is a fine-grained bird species dataset with 200 categories and 11,788 images. We follow the split of 100/50/50 classes for training/-validation/testing. **Stanford-Cars** contains 196 car categories with 16,185 images, covering different models and production years. We adopt the split of 130 training classes, 17 validation classes, and 49 test classes. **Stanford-Dogs** is a fine-grained dog breed recognition dataset with 120 categories and 20,580 images. We split the classes into 70/20/30 for training/validation/testing.

**Cross-Domain Few-Shot Classification.** **Places** is a large-scale scene recognition dataset. We follow the split of 183 training classes, 91 validation classes, and 91 test classes, with 73,000 images in total. This dataset evaluates the ability to generalize from object-centric training sets to scene-centric categories. **ChestX** is a medical imaging dataset with 25,847 chest X-ray images across 7 categories. Unlike natural images, ChestX poses significant domain shift challenges, making it a rigorous benchmark for testing cross-domain generalization.

Table 9: Performance comparison with different FSL branch on miniImageNet 1-shot tasks.

| Backbone | Params (M) | Strategy | Acc (%) | Training (min) | Memory (GB) |
|---|---|---|---|---|---|
| Visformer-T | 10.0 | Meta-tuning | 81.69 | **1.2** | **4.3** |
| CLIP ViT-B/16 | 86.7 | LoRA | 83.10 | 11.6 | 23.9 |

## A.3 COMPARISON WITH CLIP-BASED METHODS

The CLIP-based FSL line focuses on adapting a powerful vision-language model pretrained on hundreds of millions of image-text pairs, typically through prompt tuning (e.g., CoCoOp [1]) or lightweight adapters (e.g., LDC [2]). In these approaches, the CLIP backbone is fixed or lightly updated, and the performance gains largely stem from leveraging the extensive semantic prior encoded in large-scale CLIP pretraining. Additionally, their tuning scale and procedure closely resemble standard ImageNet-1k classification. In contrast, the proposed method follows the standard episodic FSL paradigm. The backbone is trained from scratch on tiny-scale few-shot datasets such as miniImageNet, where training and test classes are strictly disjoint. As a result, the goals, assumptions, and data regimes of the two FSL settings differ fundamentally.

Additionally, the proposed DVLA-RL framework is applied on CLIP vision encoder. After semantics are generated offline, DVLA-RL performs cross-modal alignment during visual feature extraction. The CLIP vision encoder processes visual tokens at each Transformer layer that are structurally consistent with our ViT-based backbones. Therefore, DVLA-RL can be seamlessly integrated without architectural modification. To further verify this, an additional experiment is conducted on miniImageNet 1-shot tasks using CLIP ViT-B/16 vision encoder. As shown in the Table below, DVLA-RL yields consistent improvements (e.g., +1.4% with LoRA tuning), confirming that the proposed framework can also enhance CLIP-based few-shot learners. However, CLIP-based training results in an increase of over five times in both time and memory usage due to the large-scale CLIP Transformer, introducing significantly higher computational overhead. This goes beyond the scope of the traditional FSL setting.

Table 10: Ablation study of different semantics.

| Semantics | miniImageNet | | CIFAR-FS | |
|---|---|---|---|---|
| | 1-shot | 5-shot | 1-shot | 5-shot |
| Name (Chen et al., 2023) | $78.96 \pm 0.31$ | $85.66 \pm 0.27$ | $85.31 \pm 0.41$ | $88.31 \pm 0.28$ |
| Definition (Zhang et al., 2024a) | $80.66 \pm 0.32$ | $87.09 \pm 0.26$ | $86.02 \pm 0.39$ | $90.08 \pm 0.30$ |
| Attribute (Liu et al., 2025) | $81.01 \pm 0.30$ | $86.81 \pm 0.26$ | $86.53 \pm 0.40$ | $89.71 \pm 0.29$ |
| Ours | $\mathbf{81.69 \pm 0.36}$ | $\mathbf{88.25 \pm 0.28}$ | $\mathbf{87.18 \pm 0.40}$ | $\mathbf{90.59 \pm 0.31}$ |

## A.4 MORE DISCUSSION ON VISION-LANGUAGE ALIGNMENT

Prior semantic-based FSL methods typically fuse text only at the classifier or prototype layers without cross-modal interaction during feature extraction, resulting in largely unaligned features between modalities. Even recent works that introduce more text (e.g., attributes (Liu et al., 2025), or descriptions (Zhang et al., 2024a)) into intermediate layers (Chen et al., 2023; Liu et al., 2025; Zhang et al., 2024a) still (i) rely on single-level semantics and (ii) apply static MLP fusion across all depths. This static and layer-agnostic design fails to account for the hierarchical nature of visual features where shallow layers capture fine local details while deeper layers focus on global abstract context, resulting in weak cross-modal alignment.

In the DVLA-RL, structured textual semantics act as an inductive prior that constrains the visual feature extractor at the appropriate level. Fine-grained attributes correspond naturally to low-level patterns (e.g., texture, color, local shape), whereas holistic descriptions correspond to high-level conceptual context. Introducing semantics aligned with the visual hierarchy provides the model with an additional structural prior, enabling each layer to attend to semantically meaningful visual evidence. To achieve adaptive cross-modal alignment across network layers, the RLA module formulates cross-modal fusion as a sequential decision process, allowing the model to dynamically emphasize self-attention or cross-attention based on the layer's semantic relevance. As a result, shallow layers capture fine local details while deeper layers focus on global abstract context, enabling more precise cross-modal alignment.

The empirical results in the manuscript further support this observation. As shown in Table 4 below, a single-layer semantics of either attributes or descriptions yields limited benefits. Furthermore, as shown in Table 5a below, static MLP-based fusion in SP, SemFew, and ECER with identical text consistently underperforms. In contrast, the best results of our DVLA-RL demonstrate that dual-level semantic structure and adaptive cross-modal alignment are crucial for effectively bridging the modality gap.

## A.5 MORE DISCUSSION ON LLM GENERATION QUALITY

The proposed DVLA-RL is designed to effectively address the correctness of LLM-generated semantics. First, DSC conditions the LLM on both class names and support images to generate more precise and visually grounded semantics, rather than on class names alone as in prior works. This eliminates the common failure mode where the LLM imagines text inconsistent with the actual image object, typically requiring expensive manual (Zhang et al., 2024a) or LLM-based pretraining (Liu et al., 2025) corrections. Second, DSC is designed to be error-tolerant. The progressive Top-k filtering stage explicitly removes noisy or inconsistent attributes. The subsequent abstraction step

Table 11: Radiologist verification of representative LLM-generated attributes on four ChestX classes.

| Class | Attributes (abbrev.) | Expert evaluation | Discriminativeness |
|---|---|---|---|
| Atelectasis | Reduced lung volume; mediastinal shift toward the lesion; elevated ipsilateral diaphragm; displaced or stretched fissure; focal increased opacity. | All attributes are textbook radiographic signs of atelectasis and clinically valid. | Moderate (can still overlap with effusion or consolidation). |
| Effusion | Blunting of the costophrenic angle; meniscus-shaped fluid level; homogeneous increased density in the lower lung field; diaphragmatic elevation; possible contralateral mediastinal shift. | All findings are accurate for pleural effusion; no incorrect or fabricated concepts observed. | Moderate (correct but sometimes subtle on frontal chest X-rays). |
| Infiltration | Patchy or ill-defined opacities; blurred lesion margins; asymmetric density changes; air bronchogram sign; possible rapid temporal progression or absorption. | Consistent with standard descriptions of pulmonary infiltration; no hallucinated terminology. | Low (semantic patterns shared with pneumonia, edema and other parenchymal diseases). |
| Nodule | Lesion < 3 cm; smooth or spiculated margins; benign or malignant calcification patterns; solitary or multiple distribution; peripheral predominance. | All attributes match clinical diagnostic criteria; no hallucination detected. | Low (substantial overlap with mass, granulomas, and other focal lesions). |
| Pneumothorax | Sharp pleural line; absence of lung markings beyond the line; increased translucency of the affected side; mediastinal shift in tension pneumothorax; ipsilateral diaphragmatic depression or flattening. | All attributes are specific and correct for pneumothorax; confirmed by the radiologist. | High (most distinctive semantic pattern among ChestX classes). |

further consolidates the remaining semantics into a coherent class-level description, making the representation less sensitive to isolated attribute errors. Third, the RLA is the adaptive alignment mechanism. The RL gate dynamically reduces unreliable textual tokens and relies more heavily on visual evidence, further preventing incorrect semantics from dominating the fusion process.

These robustness properties are validated through empirical studies in the manuscript. (1) As shown in Fig 3, removing Top-k filtering introduces significant semantic noise and notably degrades performance. (2) Table 7 further shows that DSC exhibits robustness across different LLMs, consistently achieving superior performance. In addition, we compare different forms of textual semantics under the same setting as shown in Table 10. Class name-based semantics yield the lowest performance due to minimal semantic guidance. SemFew definitions and ECER attributes offer richer coarse and fine-grained semantics but still suffer from lack of visual grounding and semantic structure. In contrast, the strong results of DSC confirm the effectiveness of the prevention and the mitigation of semantic errors.

Table 12: Ablation of the Beta concentration $\kappa$ and RL weight $\lambda$

(a) Beta concentration $\kappa$

| Value | miniImageNet | | tieredImageNet | |
|---|---|---|---|---|
| | 1-shot | 5-shot | 1-shot | 5-shot |
| 5 | 81.69 | **88.25** | **83.02** | 91.71 |
| 10 | **81.77** | 88.17 | 82.91 | 91.63 |
| 15 | 81.63 | 88.19 | 82.96 | 91.68 |
| 20 | 81.58 | 88.14 | 82.89 | **91.76** |

(b) RL weight $\lambda$

| Value | miniImageNet | | tieredImageNet | |
|---|---|---|---|---|
| | 1-shot | 5-shot | 1-shot | 5-shot |
| 0.1 | **81.69** | **88.25** | **83.02** | **91.71** |
| 0.3 | 81.44 | 88.02 | 82.81 | 91.51 |
| 0.5 | 81.32 | 88.09 | 82.50 | 91.46 |
| 1.0 | 81.08 | 89.78 | 82.41 | 91.30 |

A.6 Hyperparameter Analysis in RLA

To fully explore tuning ranges and sensitivity of hyperparameters in RLA, we additionally performed a complete ablation over Beta concentration $\kappa$ and RL weight $\lambda$. For the Beta concentration, we evaluate $\kappa \in \{5, 10, 15, 20\}$, and the results show that performance variations remain within 0.2%, indicating that DVLA-RL is highly insensitive to $\kappa$. For the RL weight, we sweep $\lambda \in \{0.1, 0.3, 0.5, 1.0\}$; $\lambda = 0.1$ yields the best accuracy, while larger values slightly degrade performance because the RL

reward begins to dominate the supervised alignment signal. These results confirm that DVLA-RL is robust to both hyperparameters and does not rely on delicate tuning.

## A.7 MORE DISCUSSION ON CROSS-DOMAIN CHESTX

To assess whether the limited gains on ChestX arise from hallucinated semantics, an additional radiologist audit of the LLM-generated attributes is conducted as shown in the Table below. The expert confirmed that all attributes correspond to clinically accurate radiographic signs, with no fabricated or incorrect concepts. Therefore, the factor that truly limits performance on ChestX is not semantic hallucination, but the intrinsic characteristics of thoracic imaging and the severe cross-domain shift. Many ChestX categories (e.g., effusion, consolidation, infiltration, edema) exhibit highly overlapping radiographic manifestations, which inherently restrict the discriminative power that semantic cues can provide. Moreover, chest X-rays are grayscale, low-texture, and anatomy-constrained, differing substantially from natural-image distributions used during training. As a result, although DVLA-RL produces clinically correct semantics, the separable semantic space in this domain is much narrower than in datasets such as CUB or Places.

Despite these challenges, DVLA-RL still achieves state-of-the-art results on ChestX, obtaining 23.47% (1-shot) and 26.94% (5-shot), which exceed the competitive MEFP method [1] by 0.7% and 0.4%, respectively. This indicates that DVLA-RL avoids the negative transfer commonly observed in medical cross-domain settings and remains one of the most robust approaches even without any medical-domain pretraining.

