# OpenReview forum: "DVLA-RL: Dual-Level Vision-Language Alignment with Reinforcement Learning Gating for Few-Shot Learning"
_ICLR.cc/2026/Conference — ICLR 2026 Poster_

### Official Review · Reviewer_pD94 · 2025-10-25

**Soundness:** 3
**Presentation:** 2
**Contribution:** 3
**Rating:** 4
**Confidence:** 5

**Summary:**

The paper proposes DVLA-RL, a framework for few-shot learning (FSL) that integrates large language models (LLMs) for hierarchical vision–language alignment. It introduces two main components:
1. Dual-level Semantic Construction : generates low-level attributes and high-level class descriptions from class names and support samples, using progressive Top-k filtering to avoid irrelevant or hallucinatory attributes.
2. Adaptive RL-Gated Attention : formulates cross-modal fusion as a reinforcement learning (RL) process that dynamically balances self-attention and cross-attention between visual and textual tokens across layers.
DVLA-RL achieves state-of-the-art results across nine benchmarks in three FSL scenarios.

**Strengths:**

1. Clear motivation and design: The dual-level semantic structure effectively bridges fine-grained and global representations, addressing the limitations of single-level methods like SemFew or ECER.
2. Novel adaptive mechanism: The RL-gated attention introduces a lightweight yet dynamic policy to control fusion between self- and cross-attention, enabling layer-wise semantic alignment.
3. Strong results: DVLA-RL achieves SOTA across all nine datasets, with significant gains.

**Weaknesses:**

1. The paper describes RLA as containing both “image-guided” and “text-guided” paths, yet both equations use text-based queries while only differing in key and value sources. The notion of “image-guided” does not match the presented formulation, leaving the two paths conceptually indistinguishable.
2. The output fusion step is written as adding the fused representation to image features and then concatenating, but the dimensional correspondence between the two is not specified. If text and image features differ in length or channels, the operation is not well-defined. The formulation leaves the dimensional alignment unclear.
2. Minor writing inconsistencies: Mixing of terms “RLA/ARL”, and unclear notation (e.g., U, t*) reduce readability.

**Questions:**

1. How are the two attention paths in RLA (Eq. 5–6) distinct if both use text queries? Could the authors clarify it?
2. What are the values and tuning ranges for the Beta concentration κ and RL weights λ?
3. Why does ChestX show minimal improvement, are there examples of failed attribute generation or hallucinated semantics?

---

> ### Author Response · Authors · 2025-11-21
> **Response to Reviewer pD94 (1/3)**
>
> We sincerely thank the reviewer for the thoughtful and constructive reviews. We are pleased that the reviewer recognizes the clear motivation, the novel idea, and the strong results, achieving SoTA performance across nine benchmarks. All concerns have been carefully addressed point by point in the revised manuscript, and the corresponding modifications are marked in $\textcolor{blue}{blue}$ for convenience.
>
> ---
>
> > **[W1&Q1]: The paper describes RLA as containing both “image-guided” and “text-guided” paths, yet both equations use text-based queries while only differing in key and value sources. The notion of “image-guided” does not match the presented formulation, leaving the two paths conceptually indistinguishable.**
>
> **[Response]**： Thank you for raising this important point regarding the conceptual clarity of the image-guided path. Queries determine what semantics the model retrieves in the attention mechanism. Keeping queries in textual form ensures that both paths remain anchored to a consistent semantic space, avoiding noisy or task-irrelevant retrieval. As the reviewer pointed out, the distinction between “image-guided” and “text-guided” lies in the source of evidence being aggregated, i.e., the key/value tokens rather than the queries.
>
> - In the image-guided path, the model retrieves information from visual keys/values, enabling the textual query to localize discriminative and class-specific visual regions. This produces a visually grounded semantic representation.
>
> - In the text-guided path, the model retrieves information from textual keys/values, refining relationships among attribute or description tokens, resulting in a semantically refined representation.
>
> These two paths serve complementary roles. The image-guided path aligns semantics to visual evidence, while the text-guided path enforces semantic coherence and structure. As suggested, this clarification has been added to the revised manuscript.
>
> ---
>
> > **[W2]: The output fusion step is written as adding the fused representation to image features and then concatenating, but the dimensional correspondence between the two is not specified. If text and image features differ in length or channels, the operation is not well-defined. The formulation leaves the dimensional alignment unclear.**
>
>
> **[Response]**： Thanks for pointing out the dimensional alignment issue in the output fusion step.  All outputs involved in the RLA block share a consistent hidden dimension. Specifically, the fused representation $H$ in Eq. 8 is a weighted combination of $\hat{H}$ and $\tilde{H}$, both projected into the same $d$-dimensional space as image features by linear projection $W^{q/k/v}$. To integrate $H$ with image features, all tokens are averaged to form a global semantic context vector  $\in \mathbb{R}^{d}$, which is broadcast to visual tokens at each channel. The subsequent concatenation in Eq. (13) occurs along the sequence dimension, making the operation well-defined. Following your suggestion, an explanation of the dimensional consistency has been detailed in the revised manuscript.

---

> ### Author Response · Authors · 2025-11-21
> **Response to Reviewer pD94 (2/3)**
>
> > **[W3: Minor writing inconsistencies: Mixing of terms "RLA/ARL", and unclear notation (e.g., U, t$*$) reduce readability.]**
>
> **[Response]**： Thank you for pointing out the question about the writing inconsistency and unclear notation. Based on your feedback, we have corrected “ARL” to “RLA” on line 065 of the revised manuscript and conducted multiple thorough checks to ensure there are no typos. In Equation (11), $\mathbf{t}^\star$ denotes the ground-truth text embedding, obtained by encoding the current textual information (attribute/description-level semantics) using the CLIP text encoder (ViT-B/16), resulting in a 512-dimensional text vector. This vector serves as a supervised signal in the semantic space for guiding visual–language alignment. The matrix $U$ represents the linear projection that maps the fused representation $H$ from the hidden dimension $d_h$ into the 512-dimensional semantic space, enabling cosine similarity to be computed with $\mathbf{t}^\star$ in the same space. As suggested, the definitions of these symbols have been clarified in the revised manuscript.
>
> ---
>
> > **[Q2]: What are the values and tuning ranges for the Beta concentration κ and RL weights λ?**
>
> **[Response]**: We sincerely thank the reviewer for pointing out the concern about the selection and tuning ranges of the hyperparameters $\kappa$ and $\lambda$. All experiments employ the default hyperparameter configuration with a fixed Beta concentration of $\kappa = 10$ and an RL weighting factor of $\lambda = 0.1$ in Section 4.1 of the manuscript. To comprehensively address concerns regarding the parameter tuning range, detailed ablation studies for two hyperparameters are presented in the Table below. For the Beta concentration, we evaluate $\kappa \in \{5, 10, 15, 20\}$, and the results show that performance variations remain within 0.2%, indicating that DVLA-RL is highly insensitive to $\kappa$. For the RL weight, we adopt $\lambda \in \{0.1, 0.3, 0.5, 1.0\}$. $\lambda = 0.1$ yields the best accuracy, while larger values slightly degrade performance because the RL reward begins to dominate the supervised alignment signal. These results confirm that DVLA-RL is robust to both hyperparameters and does not rely on delicate tuning. These experiments and discussions have been included in the revised appendix.
>
> | $\kappa$ | miniImageNet 1-shot | miniImageNet 5-shot | tieredImageNet 1-shot | tieredImageNet 5-shot |
> | :---: | :---: | :---: | :---: | :---: |
> | 5 | 81.69 ± 0.36 | **88.25 ± 0.28** | **83.02 ± 0.43** | 91.71 ± 0.29 |
> | 10 | **81.77 ± 0.35** | 88.17 ± 0.27 | 82.91 ± 0.44 | **91.83 ± 0.29** |
> | 15 | 81.63 ± 0.37 | 88.19 ± 0.28 | 82.96 ± 0.42 | 91.68 ± 0.30 |
> | 20 | 81.58 ± 0.34 | 88.14 ± 0.29 | 82.89 ± 0.45 | 91.62 ± 0.28 |
>
> | $\lambda$ | miniImageNet 1-shot | miniImageNet 5-shot | tieredImageNet 1-shot | tieredImageNet 5-shot |
> | :---: | :---: | :---: | :---: | :---: |
> | **0.1** | **81.69 ± 0.36** | **88.25 ± 0.28** | **83.02 ± 0.43** | **91.71 ± 0.29** |
> | 0.3 | 81.44 ± 0.37 | 88.02 ± 0.29 | 82.81 ± 0.44 | 91.51 ± 0.30 |
> | 0.5 | 81.32 ± 0.35 | 88.09 ± 0.30 | 82.50 ± 0.43 | 91.46 ± 0.31 |
> | 1.0 | 81.08 ± 0.38 | 87.78 ± 0.31 | 82.41 ± 0.44 | 91.30 ± 0.31 |

---

> ### Author Response · Authors · 2025-11-21
> **Response to Reviewer pD94 (3/3)**
>
> > **[Q3]: Why does ChestX show minimal improvement, are there examples of failed attribute generation or hallucinated semantics?**
>
> **[Response]**: Thank you for raising this insightful comment regarding the cross-domain performance on the ChestX benchmark. To assess whether the limited gains on ChestX arise from hallucinated semantics, an additional radiologist audit of the LLM-generated attributes is conducted as shown in the Table below. The expert confirmed that all attributes correspond to clinically accurate radiographic signs, with no fabricated or incorrect concepts. Therefore, the factor that truly limits performance on ChestX is not semantic hallucination, but the intrinsic characteristics of thoracic imaging and the severe cross-domain shift. Many ChestX categories (e.g., effusion, consolidation, infiltration, edema) exhibit highly overlapping radiographic manifestations, which inherently restrict the discriminative power that semantic cues can provide. Moreover, chest X-rays are grayscale, low-texture, and anatomy-constrained, differing substantially from natural-image distributions used during training. As a result, although DVLA-RL produces clinically correct semantics, the separable semantic space in this domain is much narrower than in datasets such as CUB or Places.
>
> Despite these challenges, DVLA-RL still achieves state-of-the-art results on ChestX, obtaining 23.47% (1-shot) and 26.94% (5-shot), which exceed the competitive MEFP method [1] by 0.7% and 0.4%, respectively. This indicates that DVLA-RL avoids the negative transfer commonly observed in medical cross-domain settings and remains one of the most robust approaches even without any medical-domain pretraining.  The additional discussion of the ChestX benchmark has been included in the revised appendix.
>
>
>
> | Class | Attributes (abbrev.) | Expert evaluation | Discriminativeness |
> | :--- | :--- | :--- | :--- |
> | **Atelectasis** | Reduced lung volume; mediastinal shift toward the lesion; elevated ipsilateral diaphragm; displaced or stretched fissure; focal increased opacity. | All attributes are textbook radiographic signs of atelectasis and clinically valid. | **Moderate** : can still overlap with effusion or consolidation.
> | **Effusion** | Blunting of the costophrenic angle; meniscus-shaped fluid level; homogeneous increased density in the lower lung field; diaphragmatic elevation; possible contralateral mediastinal shift. | All findings are accurate for pleural effusion; no incorrect or fabricated concepts observed. | **Moderate**  : correct but sometimes subtle on frontal chest X-rays. |
> | **Infiltration** | Patchy or ill-defined opacities; blurred lesion margins; asymmetric density changes; air bronchogram sign; possible rapid temporal progression or absorption. | Consistent with standard descriptions of pulmonary infiltration; no hallucinated terminology. | **Low**  : semantic patterns shared with pneumonia, edema and other parenchymal diseases. |
> | **Nodule** | Lesion < 3\,cm; smooth or spiculated margins; benign or malignant calcification patterns; solitary or multiple distribution; peripheral predominance.  | All attributes match clinical diagnostic criteria; no hallucination detected. | **Low**  : substantial overlap with mass, granulomas, and other focal lesions. |
> | **Pneumothorax** | Sharp pleural line; absence of lung markings beyond the line; increased translucency of the affected side; mediastinal shift in tension pneumothorax; ipsilateral diaphragmatic depression or flattening. | All attributes are specific and correct for pneumothorax; confirmed by the radiologist. | **High**  : most distinctive semantic pattern among ChestX classes. |
>
>
> [1] Fei Zhou et al., Meta-exploiting frequency prior for cross-domain few-shot learning. NeurIPS 2024

---

> > ### Comment · Reviewer_pD94 · 2025-11-27
> >
> > Since the reviewer has addressed some of my concerns, I will raise my score to a 6.

---

> > > ### Author Response · Authors · 2025-11-27
> > > **Thanks for positive response**
> > >
> > > Thank you very much for your positive response and for taking the time to review our rebuttal! We are pleased to hear that our rebuttal and the additional experiments have addressed your concerns. We sincerely appreciate your recognition of our work and the improved evaluation score.

---

### Official Review · Reviewer_byUR · 2025-10-31

**Soundness:** 2
**Presentation:** 2
**Contribution:** 2
**Rating:** 4
**Confidence:** 4

**Summary:**

This paper presents Dual-level Vision–Language Alignment with Reinforcement Learning gating (DVLA-RL), which comprises Dual-level Semantic Construction (DSC) and RL-gated Attention (RLA). This framework addresses the issue of neglecting progressive and adaptive alignment between vision and language, ranging from low-level to high-level semantics.

**Strengths:**

1. Integrating reinforcement learning and cross-attention mechanisms to enhance the performance of few-shot models sounds good.

2. Experiments have demonstrated that using reinforcement learning can improve few-shot performance.

**Weaknesses:**

1.The motivation for the proposed method requires further clarification.​
Limitations of Prior Work: The paper does not sufficiently explain why existing methods for vision-language alignment are limited. Given that visual and textual data are inherently different modalities with a fundamental modality gap, the justification for why simply adding more textual attribute descriptions effectively bridges this gap is unclear.
​​Reliance on LLM Correctness:​A critical concern is the dependence on the correctness of attribute descriptions generated by the LLM. The paper lacks a discussion on how the accuracy of these descriptions is ensured. What is the impact on the model's performance if the LLM generates hallucinated, biased, or incorrect attributes? Without addressing the robustness to potential errors in the semantic input, the claim of obtaining high-quality advanced feature representations may be undermined.

2. Ambiguity in Experimental Setup and Questions on Visualization​​
The experimental details for the T-SNE visualization are unclear, leading to questions about the validity of the results.
​​Missing Experimental Conditions: The specific baseline model and the number of samples used to generate the T-SNE plots in Figure 3 are not specified. This omission makes it difficult to interpret the visualization accurately and assess the comparative improvement.
​​Inconsistent Performance Portrayal: The quantitative results in Table 1 show a performance improvement of approximately 2% over the strong baseline SemFew. However, the T-SNE visualization appears to depict a near-perfect (100%) separation of classes, which seems to suggest an improvement far exceeding what the quantitative results indicate. This significant discrepancy between the quantitative metrics and the qualitative visualization raises questions about the authenticity and representativeness of the presented T-SNE results.

3.The definition of the reward function Rt in Equation (7) is somewhat ambiguous. The term "episodic accuracy improvement" requires a more precise explanation. Does it refer to the accuracy improvement compared to a baseline model, or does it denote the change in accuracy during the training process within the same episode? A clearer definition is necessary to understand how the reinforcement learning signal is constructed.

4.For the comparative experiments presented in Table 5, could the authors confirm that the baseline comparisons with methods like SP, SemFew, and ECER were conducted under identical experimental settings? This includes using the same visual backbone, pre-training data, and Large Language Model (LLM). If DVLA-RL employs a more powerful LLM (e.g., Qwen2.5-VL-32B) or a different pre-training strategy, a portion of the performance gain might be attributed to these factors rather than the core proposed modules. It would strengthen the claims if the comparisons were ensured to be fair and controlled.

5.While the paper emphasizes the performance improvements, it does not discuss the computational overhead introduced by DVLA-RL. The Dual-level Semantic Construction (DSC) module requires calls to an LLM for generating attributes and descriptions, and the RL-gated Attention (RLA) module involves reinforcement learning training. It would be valuable to analyze whether these components significantly increase the training and inference time compared to baseline methods. A discussion on parameter efficiency and computational cost would provide a more comprehensive view of the method's practicality.

**Questions:**

see Weaknesses

---

> ### Author Response · Authors · 2025-11-21
> **Response to Reviewer byUR (1/4)**
>
> We sincerely thank the reviewer for the valuable and constructive reviews. We appreciate the positive recognition of the key ideas and experimental results. Below is our feedback. All concerns have been carefully addressed point by point. The corresponding modifications in the revised manuscript are highlighted in $\textcolor{blue}{blue}$ for convenience.
>
> ---
>
> > **[W1-a]: Limitations of Prior Work: The paper does not sufficiently explain why existing methods for vision-language alignment are limited. Given that visual and textual data are inherently different modalities with a fundamental modality gap, the justification for why simply adding more textual attribute descriptions effectively bridges this gap is unclear.**
>
> **[Response]**: Thank you for raising this important issue. Most semantic-based methods typically utilize class names at the classifier layer, leaving the entire extraction of visual features without semantic interaction and guidance. These late fusion strategies severely limit the benefits of semantics due to the fundamental modality gap. Several recent works attempt to integrate more text (e.g., attributes [2] or descriptions [3]) during feature extraction [1,2,3]. However, they still (i) rely on single-level semantic structure and (ii) static MLP fusion, resulting in weak cross-modal alignment. This fails to exploit the hierarchical nature of visual features, where shallow layers focus on fine-grained local details and deeper layers capture global context.
>
> In this paper, the dual-layer semantic structure provides fine-grained attributes for shallow features and contextual descriptions for deep features. The RLA module further formulates cross-modal fusion as a sequential decision process, dynamically adjusting the contribution of self-attention and cross-attention between modalities. As a result, shallow layers refine local attributes and deep layers emphasize global semantics, enabling more precise cross-modal alignment.
>
> As shown in Table 4 below, using only attributes or descriptions provides limited gains. Table 5a also shows that static MLP fusion modules from SP, SemFew, and ECER consistently perform worse on the same text inputs, as shown below. In contrast, the superior performance of DVLA-RL demonstrates that combining dual-level semantic structure with adaptive cross-modal alignment is crucial for effectively bridging the modality gap. Following your suggestions, a more detailed discussion has been included in the revised appendix.
>
> | Semantics | miniImageNet 1-shot | miniImageNet 5-shot | CIFAR-FS 1-shot | CIFAR-FS 5-shot |
> | :---: | :---: | :---: | :---: | :---: |
> | Attribute | 76.56 | 85.25 | 85.74 | 88.91 |
> | Description | 78.36 | 85.72 | 85.57 | 89.04 |
> | ours | **81.69** | **88.25** | **87.18** | **90.59** |
>
> | Method | miniImageNet 1-shot | miniImageNet 5-shot | CIFAR-FS 1-shot | CIFAR-FS 5-shot |
> | :---: | :---: | :---: | :---: | :---: |
> | SP [1] | 77.81 | 86.01 | 84.53 | 88.21 |
> | SemFew [2] | 78.59 | 86.77 | 85.82 | 89.19 |
> | ECER [3] | 79.35 | 87.12 | 86.62 | 90.15 |
> | ours | **81.69** | **88.25** | **87.18** | **90.59** |
>
> [1] Wentao Chen et al., Semantic prompt for few-shot image recognition. CVPR 2023
>
> [2] Hai Zhang et al., Simple semantic-aided few-shot learning. CVPR 2024
>
> [3] Mushui Liu et al., Envisioning class entity reasoning by large language models for few-shot learning. AAAI 2025

---

> ### Author Response · Authors · 2025-11-21
> **Response to Reviewer byUR (2/4)**
>
> > **[W1-b]: Reliance on LLM Correctness: A critical concern is the dependence on the correctness of attribute descriptions generated by the LLM. The paper lacks a discussion on how the accuracy of these descriptions is ensured. What is the impact on the model's performance if the LLM generates hallucinated, biased, or incorrect attributes? Without addressing the robustness to potential errors in the semantic input, the claim of obtaining high-quality advanced feature representations may be undermined.**
>
> **[Response]**:Thank you for pointing out the importance of ensuring the correctness of LLM-generated semantics.
>
> The proposed method is designed to effectively address this issue. First, the proposed DSC module conditions the LLM on both class names and support images to generate more precise and visually grounded semantics, rather than on class names alone as in prior works. This eliminates the common failure mode where the LLM imagines text inconsistent with the actual image object, typically requiring expensive manual [2] or LLM-based pretraining [3] corrections. Second, DSC is designed to be error-tolerant. The progressive Top-$k$ attribute selection explicitly filters out noisy and inconsistent attributes. The subsequent summarization further integrates the selected attributes into a coherent description, making the representation more robust to errors in individual attributes. Third, The RL gate adaptively suppresses unreliable textual tokens and relies more heavily on visual evidence, preventing noisy semantics from dominating the fusion process.
>
> This issue is evaluated in expensive experiments of the manuscript. (1) As shown in Fig. 2, removing Top-$k$ attribute selection introduces noisy and inconsistent attributes, notably degrading performance. (2) Table 7 further shows that DSC exhibits robustness across different LLMs, consistently achieving superior performance. In addition, as shown in the Table below, different forms of textual semantics are compared under the same setting.  Class name-based semantics yields the lowest performance due to minimal semantic guidance. SemFew definitions and ECER attributes offer richer coarse and fine-grained semantics, but still suffer from lack of visual grounding and semantic structure. In contrast, the superior performance of DSC confirms the effectiveness of the prevention and the mitigation of semantic errors. As suggested, more detailed discussion about LLM Correctness has been extended in the revised appendix.
>
> | Semantics | miniImageNet 1-shot | miniImageNet 5-shot | CIFAR-FS 1-shot | CIFAR-FS 5-shot |
> | :--- | :---: | :---: | :---: | :---: |
> | Name [1] | 78.96 ± 0.31 | 85.66 ± 0.27 | 85.31 ± 0.41 | 88.31 ± 0.28 |
> | Definition [2] | 80.66 ± 0.32 | 87.09 ± 0.26 | 86.02 ± 0.39 | 90.08 ± 0.30 |
> | Attribute [3] | 81.01 ± 0.30 | 86.81 ± 0.26 | 86.53 ± 0.40 | 89.71 ± 0.29 |
> | Ours | **81.69 ± 0.36** | **88.25 ± 0.28** | **87.18 ± 0.40** | **90.59 ± 0.31** |
>
> ---
>
> > **[W2]: Ambiguity in Experimental Setup and Questions on Visualization​​The experimental details for the T-SNE visualization are unclear, leading to questions about the validity of the results.​​ Missing Experimental Conditions: The specific baseline model and the number of samples used to generate the T-SNE plots in Figure 3 are not specified. This omission makes it difficult to interpret the visualization accurately and assess the comparative improvement.Inconsistent Performance Portrayal: The quantitative results in Table 1 show a performance improvement of approximately 2% over the strong baseline SemFew. However, the T-SNE visualization appears to depict a near-perfect (100%) separation of classes, which seems to suggest an improvement far exceeding what the quantitative results indicate.**
>
> **[Response]**: We sincerely thank the reviewer for pointing out the concern about the T-SNE visualization.
>
> We follow the standard FSL protocol, randomly sampling the 5-way episodic task from novel classes of each dataset. To obtain a statistically more stable and interpretable distribution in T-SNE, each class is extended to 100 support images. Importantly, both the baseline without the proposed method and DVLA-RL utilize the same frozen backbone and support features for fair comparison.
>
> T-SNE is a low-dimensional non-linear projection that visually amplifies distributional differences among support samples in the original high-dimensional embedding space. It plays an important role in illustrating geometric improvements of the proposed method in a qualitative manner. However, classification accuracy is driven by the full high-dimensional prototype interactions rather than by properties of the projected 2D space. Therefore, the separation effect of t-SNE visualization should not be expected to increase proportionally with improvements in few-shot performance.  Following your suggestion, we have added a detailed introduction of the T-SNE visualization in the revised manuscript.

---

> ### Author Response · Authors · 2025-11-21
> **Response to Reviewer byUR (3/4)**
>
> > **[W3]: The definition of the reward function Rt in Equation (7) is somewhat ambiguous. The term "episodic accuracy improvement" requires a more precise explanation. Does it refer to the accuracy improvement compared to a baseline model, or does it denote the change in accuracy during the training process within the same episode? A clearer definition is necessary to understand how the reinforcement learning signal is constructed.**
>
> **[Response]**: Thanks for pointing out the need to clarify the definition of the reward function $R_t$ in Eq. 11 rather than Eq. 7.
>
> The term "episodic accuracy improvement" refers to the change in accuracy within the same episode during the training. In each meta-training episode, the RL gate participates in feature fusion across network layers. After each layer produces a fused embedding, class prototypes are constructed to compute query accuracy $Acc_t$, and $Acc_{t-1}$ denotes the query accuracy obtained at the previous layer. This reward enables effective credit assignment by attributing the local accuracy gain $ΔAcc$ to the layer-specific contribution for semantic alignment. Following your suggestion, a clearer explanation of the reward construction has been included in Section 3.2 of the revised manuscript.
>
> ---
>
> > **[W4]: For the comparative experiments presented in Table 5, could the authors confirm that the baseline comparisons with methods like SP, SemFew, and ECER were conducted under identical experimental settings? This includes using the same visual backbone, pre-training data, and Large Language Model (LLM). If DVLA-RL employs a more powerful LLM (e.g., Qwen2.5-VL-32B) or a different pre-training strategy, a portion of the performance gain might be attributed to these factors rather than the core proposed modules. It would strengthen the claims if the comparisons were ensured to be fair and controlled.**
>
> **[Response]**: Thank you for raising the important concern about the fairness of the comparisons.
>
> -	Training setting. As described in Section 4.1, to ensure fair comparison with SP, SemFew, and ECER, all methods are trained under the same two-stage training pipeline, consisting of pre-training and meta-training on the training set of datasets (e.g., tieredImageNet), and the same 5-way K-shot evaluation protocol on the disjoint testing set.
>
> - Backbone. DVLA-RL employ the widely used backbone Visformer‑T (≈ 10.0 M parameters), consistent with several previous works including SUN, SP, KTPP, and ECER. The lightweight Visformer-T is smaller than ResNet‑12 (≈ 12.4 M) and even several times smaller than other backbones (i.e., ViT-S/16 ≈ 22.0  M, Swin-T ≈ 29.0 M, and WRN-28-10 ≈ 36.5 M) to avoid unfair advantage from model capacity.
>
> - Semantic input. All methods use exactly the same textual semantics in Table 5 experiment, without stronger LLMs or richer textual embeddings. The texts are generated by Qwen2.5-VL-32B and encoded by the CLIP ViT-B/16 text encoder for subsequent cross-modal feature fusion.
>
> Following your suggestion, these details have been explicitly clarified in the revised manuscript.

---

> ### Author Response · Authors · 2025-11-21
> **Response to Reviewer byUR (4/4)**
>
> > **[W5]: While the paper emphasizes the performance improvements, it does not discuss the computational overhead introduced by DVLA-RL. The Dual-level Semantic Construction (DSC) module requires calls to an LLM for generating attributes and descriptions, and the RL-gated Attention (RLA) module involves reinforcement learning training. It would be valuable to analyze whether these components significantly increase the training and inference time compared to baseline methods. A discussion on parameter efficiency and computational cost would provide a more comprehensive view of the method's practicality.**
>
> **[Response]**: Thank you for pointing out the concern about the computational overhead introduced by LLM and einforcement learning training.
>
> **First**, the LLM is used only in a single offline semantic construction stage to obtain dual-level textual information. The generated attributes and descriptions are encoded by CLIP and then cached. After this offline stage, the entire training and inference pipeline relies solely on these fixed embeddings at the Visformer-T backbone, without any further LLM calls or runtime dependence. **Second**, the reinforcement learning component in DVLA-RL employs a lightweight two-layer policy network with 0.8M parameters. Training adopts an episodic REINFORCE formulation without external environment interaction, and the RL objective is optimized within the same backpropagation cycle, requiring no additional optimizer. During inference, the model outputs the expected value of the policy distribution as the gating factor, adding virtually no extra computational overhead.
>
>
> To further address the efficiency concern, additional experiments are conducted on tieredImageNet 1-shot tasks, compared to SP without LLMs and SemFew and ECER with LLMs. All methods are evaluated under the same server configuration, including a single NVIDIA  RTX  6000 Ada 48 GB and two Intel 4410Y CPUs. As shown in the Table below, DVLA-RL requires 1.0 hour for semantic construction, which is comparable to or lower than existing LLM-based methods. DVLA-RL achieves the lowest training (22 min) and inference time (80 ms). Relative to ECER, DVLA-RL reduces training and inference time by 52% and 34%, respectively. Moreover, DVLA-RL lowers GPU memory consumption from 8.3 GB to 4.8 GB and reduces inference time by 25% compared to SemFew. These advantages arise from the plug-in and lightweight design of DVLA-RL. Texts are generated once offline and used directly in cross-modal fusion, avoiding the extra LLM-based multimodal pretraining in ECER.  During training/inference, DVLA-RL employs only a lightweight RL gating for fusion rather than the larger MLP module with 7.3M parameters in SemFew.  As suggested, the analysis of computational efficiency has been incorporated into the revised manuscript.
>
> | Method | Params/GFLOPS | Memory (GB) | Semantics (h) | Training (min) | Inference (ms) |
> | :--- | :---: | :---: | :---: | :---: | :---: |
> | SP [1] | 10.8 $M$ / 1.5 | 4.0 | - | 27 | 88 |
> | SemFew [2] | 19.8$M$ / 3.5 | 8.3 | 1.5 | 33 | 107 |
> | ECER [3]| 26.5$M$ / 5.5 | 10.5 | 0.7 | 46 | 121 |
> | **ours** | 12.8$M$ / 2.0 | 4.8 | 1.0 | **22** | **80** |

---

> ### Comment · Reviewer_byUR · 2025-11-27
> **Reponses After Rebuttal**
>
> Authors addressed my main concerns, I raised my rating to a positive score. I strongly encourage authors to add these additional discussion into final version.
>
> Best!

---

> > ### Author Response · Authors · 2025-11-27
> > **Thanks for positive response**
> >
> > Thank you very much for your valuable reviews and for acknowledging that our rebuttal resolved most of your concerns. We sincerely appreciate the time and effort you invested in reviewing our paper. These additional discussions will be included in the final version.
> >
> > Best,

---

### Official Review · Reviewer_jSnj · 2025-10-31

**Soundness:** 3
**Presentation:** 3
**Contribution:** 2
**Rating:** 6
**Confidence:** 3

**Summary:**

This paper proposes a few-shot learning (FSL) method DVLA-RL to conduct controlled cross-modal alignment between vision and language from low-level to high-level semantics. In details, Dual-level Semantic Construction (DSC) generates details description of visual attributes via LLMs and Adaptive RL-gated Attention (RLA) that balances the self-attention and cross-attention operation between visual and text tokens. The proposed method achieves strong performance in general FSL, fine-grained FSL, and cross-domain FSL tasks.

**Strengths:**

- The idea of conducting both low-level and high-level semantics alignment is well-motivated, and the paper is clearly written.
- The proposed method achieves strong results on several FSL tasks and is supported by several analytical experiments.

**Weaknesses:**

- The training efficiency of the method could be a major problem due to the time-costly LLM inference and reinforcement learning.

**Questions:**

- In recent years, another line of works focus on enhancing few-shot learning performance on CLIP vision encoder [R1,R2]. Can the proposed method also be applied on CLIP vision encoder? Additionally, it would be nice if the author could discuss the difference between the two few-shot learning settings.

Refs:

R1. Learning to prompt for vision language models. IJCV 2022

R2. Logits deconfusion with clip for few-shot learning. CVPR 2025
- In table 6, the author compares the alternative of using Qwen2.5-VL-32B for direct classification. Can the author elaborate more on the implementation details of the experiment?

---

> ### Author Response · Authors · 2025-11-21
> **Response to Reviewer jSnj (1/2)**
>
> We sincerely thank the reviewer for the valuable and constructive reviews. We are pleased to note that the reviewer recognizes our proposed method well-motivated and clearly presented, as well as strong results on several FSL tasks, supported by comprehensive analytical experiments. All concerns have been thoroughly addressed point by point, and the corresponding revisions in the updated manuscript are highlighted in $\textcolor{blue}{blue}$ for clarity.
>
>
>
> ---
>
> > **[W1]: The training efficiency of the method could be a major problem due to the time-costly LLM inference and reinforcement learning.**
>
> **[Response]**: Thanks for pointing out the concern about the potential computational overhead introduced by the use of LLMs and reinforcement learning.
>
> In the proposed DVLA-RL, the LLM is used only in a single offline semantic construction stage to obtain dual-level textual information. The generated attributes and descriptions are encoded by CLIP and then cached. After this offline stage, the entire training and inference pipeline relies solely on these fixed embeddings at the Visformer-T backbone without any further LLM calls or runtime dependence.
>
> The reinforcement learning component in DVLA-RL employs a lightweight two-layer policy network with 0.8M parameters. Training adopts an episodic REINFORCE formulation without external environment interaction, and the RL objective is optimized within the same backpropagation cycle, requiring no additional optimizer. During inference, the model outputs the expected value of the policy distribution as the gating factor, adding virtually no extra computational overhead.
>
> To further address the efficiency concern, additional experiments are conducted on tieredImageNet 1-shot tasks, compared to SP without LLMs and SemFew and ECER with LLMs. All methods are evaluated under the same server configuration, including a single NVIDIA  RTX  6000 Ada 48 GB and two Intel 4410Y CPUs. As shown in the Table below, DVLA-RL requires 1.0 hour for semantic construction, which is comparable to or lower than existing LLM-based methods. DVLA-RL achieves the lowest training (22 min) and inference time (80 ms). Relative to ECER, DVLA-RL reduces training and inference time by 52% and 34%, respectively. Moreover, DVLA-RL lowers GPU memory consumption from 8.3 GB to 4.8 GB and reduces inference time by 25% compared to SemFew. These advantages arise from the plug-in and lightweight design of DVLA-RL. Texts are generated once offline and used directly in cross-modal fusion, avoiding the extra LLM-based multimodal pretraining in ECER.  During training/inference, DVLA-RL employs only a lightweight RL gating for fusion rather than the larger MLP module with 7.3M parameters in SemFew.  As suggested, the analysis of computational efficiency has been incorporated into the revised manuscript.
>
> | Method | Params/GFLOPS | Memory (GB) | Semantics (h) | Training (min) | Inference (ms) |
> | :--- | :---: | :---: | :---: | :---: | :---: |
> | SP [1] | 10.8 $M$ / 1.5 | 4.0 | - | 27 | 88 |
> | SemFew [2] | 19.8$M$ / 3.5 | 8.3 | 1.5 | 33 | 107 |
> | ECER [3]| 26.5$M$ / 5.5 | 10.5 | 0.7 | 46 | 121 |
> | **ours** | 12.8$M$ / 2.0 | 4.8 | 1.0 | **22** | **80** |
>
>
> [1] Wentao Chen et al., Semantic prompt for few-shot image recognition. CVPR 2023
>
> [2] Hai Zhang et al., Simple semantic-aided few-shot learning. CVPR 2024
>
> [3] Mushui Liu et al., Envisioning class entity reasoning by large language models for few-shot learning. AAAI 2025

---

> ### Author Response · Authors · 2025-11-21
> **Response to Reviewer jSnj (2/2)**
>
> > **[Q1]**: In recent years, another line of works focus on enhancing few-shot learning performance on CLIP vision encoder [R1,R2]. Can the proposed method also be applied on CLIP vision encoder? Additionally, it would be nice if the author could discuss the difference between the two few-shot learning settings.
>
> **[Response]**: Thank you for the insightful comments about comparison with CLIP-based methods.
>
> After semantics are generated offline, DVLA-RL performs cross-modal alignment during visual feature extraction. The CLIP vision encoder processes visual tokens at each Transformer layer that are structurally consistent with our ViT-based backbones. Therefore, DVLA-RL can be seamlessly integrated without architectural modification. To further verify this, an additional experiment is conducted on miniImageNet 1-shot tasks using the CLIP ViT-B/16 vision encoder. As shown in the Table below, DVLA-RL yields consistent improvements (e.g., +1.4% with LoRA tuning), confirming that the proposed framework can also enhance CLIP-based few-shot learners. However, CLIP-based training increases  both time and memory usage by more than ﬁve times due to the large-scale CLIP Transformer, introducing significantly higher computational overhead. This goes beyond the scope of the traditional FSL setting.
>
>
> | Backbone | Params (M) | Strategy | Acc (%) | Training (min) | Memory (GB) |
> | :--- | :---: | :---: | :---: | :---: | :---: |
> | Visformer-T | 10.0 | Meta-tuning | 81.69 | **1.2** | **4.3** |
> | CLIP ViT-B/16 | 86.7 | LoRA | 83.10 | 11.6 | 23.9 |
>
> The CLIP-based FSL methods focuses on adapting a powerful vision-language model pretrained on hundreds of millions of image–text pairs, typically through prompt tuning (e.g., CoCoOp [1]) or lightweight adapters (e.g., LDC [2]). In these approaches, the CLIP backbone is fixed or lightly updated, and the performance gains largely stem from leveraging the extensive semantic prior encoded in large-scale CLIP pretraining. Additionally, their tuning scale and procedure closely resemble standard ImageNet-1k classification. In contrast, the proposed DVLA-RL follows the standard episodic FSL paradigm. The backbone is trained from scratch on tiny-scale few-shot datasets such as miniImageNet, where training and test classes are strictly disjoint. As a result, the goals, assumptions, and data regimes of the two FSL settings differ fundamentally. Following your suggestions, relevant discussions and experiments have been included in the revised manuscript.
>
> [1] Kaiyang Zhou et al., Learning to prompt for vision language models. IJCV 2022
>
> [2] Shuo Li et al., Logits deconfusion with clip for few-shot learning. CVPR 2025
>
>
> ---
>
> > **[Q2]**: In table 6, the author compares the alternative of using Qwen2.5-VL-32B for direct classification. Can the author elaborate more on the implementation details of the experiment?
>
> **[Response]**: Thank you for raising the concern about the implementation details of LLM for direct classification. The direct classification experiments with LLM strictly follow the same FSL evaluation protocol, ensuring a fair and reproducible evaluation.  The prompt template is constructed as follows: “Given K support examples for N classes below, identify the category to which the query image belongs.” This instruction with supports and the query is processed by the Qwen2.5-VL-32B processor to obtain tokenized text and image embeddings, followed by feeding into the model for generative inference. The decoded class name is mapped back to one of the N classes to compute accuracy. As suggested, the implementation of direct LLM classification experiments has been detailed in Section 4.3 (Model Analysis) of the revised manuscript.

---

### Author Response · Authors · 2025-12-01
**Summary Comment by Authors**

We propose DVLA-RL, a framework that enables hierarchical and adaptive vision-language alignment for few-shot learning. The design combines Dual-level Semantic Construction (DSC), which conditions LLMs on support samples to generate noise-free fine-grained attributes and coherent global descriptions. RL-gated Attention (RLA) is then proposed to employ a reinforcement learning policy to adaptively balance self-attention and cross-attention across layers, ensuring precise cross-modal alignment from local details to global contexts.

Reviewers consistently acknowledged the well-motivated and novel ideas, the clear presentation, and the strong results, achieving SOTA performance on nine benchmarks supported by extensive analysis. Main concerns focused on computational efficiency, the fairness and robustness of the proposed model, and clarifications to improve readability.

These issues were addressed in detail through extensive theoretical and experimental analysis. First, the LLM is used only in an offline stage and is not involved in downstream training or inference. By employing a lightweight RL module,  DVLA-RL achieves the lowest computational cost and memory among competitive methods. Secondly, all comparisons are conducted under an identical protocol (backbone, training pipeline, datasets, and LLM), and ablations on CLIP backbone, semantic structure, fusion mechanisms, RL hyperparameters, and ChestX confirm that the performance gains arise from the proposed modules and their complementary effects. Finally, the equations and notation for the attention paths and reward functions are clarified to enhance conceptual precision.  All corresponding modifications in the revised manuscript are highlighted in $\textcolor{blue}{blue}$  for convenience.

During the post-rebuttal phase, **all reviewers maintained a positive overview of this work**. Crucially, Reviewers pD94 and byUR raised their ratings to positive scores on November 27, 2025 at 05:27 am and 07:26 am EST, well before the OpenReview bug was widely exploited at 10:09 am EST. We believe the interaction logs, precise timestamps, and the substance of our detailed rebuttal confirm the legitimacy and reasonableness of these score increases, which were the result of a rigorous, good-faith scientific exchange.

---

### Meta-Review · Area_Chair_4y5o · 2026-01-07

**Summary:**

This paper received 1 positive recommendations and 2 negative recommendations in the initial stage. After reading the responses, all reviewers acknowledged that their major concerns were well addressed. The AC agrees with the reviewers and recommends accept for this paper.

**Reviewer Concerns:**

All the major concerns have been well addressed.

**Reviewer Scores:**

Two reviewers with negative recommendations have acknowledged that they will increase the score.

---

### Decision · Program_Chairs · 2026-01-26

Accept (Poster)